# Overlapping Mechanisms of Action of Brain-Active Bacteria and Bacterial Metabolites in the Pathogenesis of Common Brain Diseases

**DOI:** 10.3390/nu14132661

**Published:** 2022-06-27

**Authors:** Tanja Patricia Eicher, M. Hasan Mohajeri

**Affiliations:** Department of Anatomy, University of Zurich, Winterthurerstrasse 190, 8057 Zurich, Switzerland; tanjapatricia.eicher@uzh.ch

**Keywords:** bacteria, metabolites, dysbiosis, therapy, microbiota–gut–brain axis, leaky gut, neuroinflammation, neurodegeneration, neurodevelopment, neuropsychiatric disorder

## Abstract

The involvement of the gut microbiota and the metabolites of colon-residing bacteria in brain disease pathogenesis has been covered in a growing number of studies, but comparative literature is scarce. To fill this gap, we explored the contribution of the microbiota–gut–brain axis to the pathophysiology of seven brain-related diseases (attention deficit hyperactivity disorder, autism spectrum disorder, schizophrenia, Alzheimer’s disease, Parkinson’s disease, major depressive disorder, and bipolar disorder). In this article, we discussed changes in bacterial abundance and the metabolic implications of these changes on disease development and progression. Our central findings indicate that, mechanistically, all seven diseases are associated with a leaky gut, neuroinflammation, and over-activated microglial cells, to which gut-residing bacteria and their metabolites are important contributors. Patients show a pro-inflammatory shift in their colon microbiota, harbouring more Gram-negative bacteria containing immune-triggering lipopolysaccharides (LPS) in their cell walls. In addition, bacteria with pro-inflammatory properties (*Alistipes*, *Eggerthella*, *Flavonifractor*) are found in higher abundances, whereas lower abundances of anti-inflammatory bacteria (*Bifidobacterium*, *Coprococcus*, *Eucbacterium*, *Eubacterium rectale*, *Faecalibacterium*, *Faecalibacterium prasunitzii*, *Lactobacillus*, *Prevotella*, *Roseburia*) are reported, when compared to healthy controls. On the metabolite level, aberrant levels of short-chain fatty acids (SCFAs) are involved in disease pathogenesis and are mostly found in lower quantities. Moreover, bacterial metabolites such as neurotransmitters (acetylcholine, dopamine, noradrenaline, GABA, glutamate, serotonin) or amino acids (phenylalanine, tryptophan) also play an important role. In the future, defined aberrations in the abundance of bacteria strains and altered bacterial metabolite levels could likely be possible markers for disease diagnostics and follow-ups. Moreover, they could help to identify novel treatment options, underlining the necessity for a deeper understanding of the microbiota–gut–brain axis.

## 1. Introduction

Neuropsychiatric diseases cover a wide spectrum of diseases affecting the brain, behaviour, and mood, affecting people of any age. Neurodevelopmental disorders such as attention deficit hyperactivity disorder and autistic disorder are prevalent in children and adolescents [1,2,3,4]. Affective disorders such as bipolar disorder or major depressive disorder mainly affect young adults in their second to third life decade [5,6]. Neurodegenerative disorders, on the other hand, are present in the elderly (60–80 years) [7,8,9,10]. Worldwide, neuropsychiatric disorders are among the most disabling diseases causing major impairments of life quality. Especially in schizophrenia, Alzheimer’s disease, and bipolar disorder, life expectancy is massively reduced by as much as 10–20 years [7,8,11,12]. This highlights the importance of a profound understanding of the pathophysiological mechanisms involved in disease development.

In recent years, the contribution of the gut microbiota and its metabolites in disease development has become the centre of attention in science. Research focusing on the microbiota–gut–brain axis has experienced a dramatic rise in recent decades, uncovering several pathways connecting the residing intestinal bacteria and different somatic, neurological, and psychiatric diseases. Disruptions in microbial compositions have been implicated in diseases such as asthma, diabetes, inflammatory bowel disease, obesity, and autism [13].

Microbial communities’ influence on the brain is complex, and the bidirectional communication between the brain and the microorganisms relies on the bacteria’s ability to produce various metabolites to interact with the hosts’ immune system and neurochemistry [14]. Many studies aiming at deciphering these interactions have been performed on animal model organisms. Only recently, more human clinical studies have been conducted, with the ultimate goal to use the newly gained knowledge to identify future treatment options [14,15].

Existing literature primarily focuses on the relationship between the intestinal microbiome and individual diseases. There is still a lack of more comparative approaches. In this review, we highlight the potential common characteristics of the involvement of the microbiota–gut–brain axis in the pathogenesis of a wide variety of neuropsychiatric diseases, covering a broad spectrum of pathologies, ranging from neurodevelopment to neurodegeneration, psychiatric and affective disorders.

## 2. Materials and Methods

The goal of this literature review is to answer the following research questions: What are potential overlapping changes in the gut microbiome in terms of bacterial strains and bacterial metabolites in people suffering from common neurodevelopmental, neurodegenerative, or mental/neuropsychiatric disorders? The discussed disorders include attention deficit hyperactivity disorder, autistic disorder, schizophrenia, Alzheimer’s disease, Parkinson’s disease, depression, and bipolar disorder. The literature research was run on the databases PubMed and Scopus until the 31 January 2022 using the following search and MeSH terms: “microbiota-gut-brain-axis”, “Gastrointestinal Microbiome” [Mesh], “Schizophrenia” [Mesh], “Alzheimer Disease” [Mesh], “Parkinson Disease” [Mesh], “Depression” [Mesh], “Bipolar Disorder” [Mesh], “Attention Deficit Disorder with Hyperactivity” [Mesh], “Autistic Disorder” [Mesh].

Within the intestinal microbiota, our primary interest was in bacteria rather than fungi, viruses, or protozoa.

The papers included had to fulfil the listed inclusion criteria below:a.Discussing bacteria or bacterial metabolites and at least one of the diseases of choice;b.Including findings from human studies with the support of preclinical data;c.Published in a peer-reviewed journal;d.Available in full-text;e.Written in English.f.Published within the time frame of January 2017–January 2022. Papers published before January 2017 were included if they were referred to in another paper.

## 3. Evidence of Linking the Microbiome–Gut–Brain Axis to Brain Disorders

### 3.1. Microbiome

The human body is densely populated with microbes, where each body site has an individual flora. The microbiota in the gut is the biggest, with approximately 10^14^ living microorganisms, including bacteria, viruses, archaea, eukaryotes, and fungi [14,16], which is equivalent to the number of human cells in the body [17]. Altogether these microorganisms are estimated to weigh around 2 kg. They encode over 232 million genes, suggesting that they may be able to exhibit a huge metabolic capacity [14,17].

In this review, “microbiome” will solely refer to the bacteria residing in the colon. Most bacteria of the intestinal microbiome are situated in the colon. Only relatively few can be found in the stomach or the small intestine due to unfavourable living conditions in these parts of the gastrointestinal (GI) tract (fast passage, gastric acid, bile, pancreas juices) [16,17]. Over the human lifespan, the microbiome continues to evolve: it first starts developing in the womb and during birth, stabilises and increases in diversity during childhood and adolescence [16,17], and is influenced by the ageing processes later on in life [15]. Furthermore, it is also highly susceptible to environmental and lifestyle factors such as drugs, antibiotics, toxins, stress, and diet [14,18,19,20], resulting in big interindividual and temporal differences [15,16,21].

Even though gut bacteria populations are highly dynamic, the mature microbiome after the age of three years is dominated by two phyla, namely Bacteroidetes and Firmicutes [15,17]. In addition, looking at the most abundant genus, three enterotypes could be distinguished: (i) Bacteroides, (ii) Prevotella, and (iii) Ruminococcus [22]. These enterotypes have been linked to different dietary patterns. Bacteroides enterotype, which has only little Pretovella, is found in individuals with Western diets (high in saturated fats, high in animal protein). Pretovella enterotype, in contrast, is present in people consuming a plant-based diet (high in fibres and carbohydrates) [21,23]. In addition to environmental influences, twin studies have shown that the composition of the microbiome may also be determined by genetic components [16,19].

A balance in the microbial composition is known as a state of eubiosis [18]. Bacteria and humans have evolved together, forming a symbiotic host–microbiome relationship [16]. Gut microbes are important for gut motility, barrier homeostasis, maintenance of gut integrity, regulation of the host immune system, absorption, and production of nutrients [13,16]. If the bacterial composition is altered, also recognised as dysbiosis, those beneficial mechanisms might be disrupted [16]. In dysbiosis, potentially harmful and inflammatory bacteria take overhand, leading to an imbalance in immune homeostasis and increased permeability in the gut. This, in turn, allows the migration of bacteria from the gut into blood circulation, which is a risk factor for systemic inflammation [18]. As a result, intestinal dysbiosis has been linked to the pathogenesis of different diseases and unfavourable health conditions such as obesity, asthma, diabetes, autism, and inflammatory bowel disease [13]. A disrupted microbiome has also been associated with different neuropsychiatric diseases, including depression, autistic disorder, Parkinson’s disease, and schizophrenia [19]. Extensive research, especially during the last decade, has been instrumental in revealing the importance of gut microbiota for host health. The question of whether dysbiosis is the underlying root cause of those diseases or the effect of the disease pathologies still needs to be understood [17].

### 3.2. Microbiota–Gut–Brain Axis (MGBA)

The microbiota–gut–brain axis (MGBA) is a bidirectional communication pathway between the gut bacteria and the central nervous system (CNS) [14,15]. It is an extension of the gut–brain axis, in which the enteric nervous system (ENS), CNS, and the GI-tract work together to affect physiological aspects of the gut: motility, secretion, and acid and mucus production [24].

Within this complex communication network (Figure 1), several different routes of interaction have been described. The bacteria may influence the brain via the production of neurotransmitters and bacterial metabolites via stimulation of the vagal nerve, the immune system, or the hypothalamic–pituitary–adrenal axis (HPA-axis) [13,14]. On the other hand, the brain’s effects on the gut in terms of secretion, peristalsis, and sensory are mainly transferred via the vagus nerve [25]. Due to the complexity of the system further work is still required to understand the details of the bidirectional communication pathways [21]. Nonetheless, we will highlight some major ways of communication here.

#### 3.2.1. Chemical Signalling

Certain types of bacteria can produce small molecules which influence the metabolism of the nervous system, either directly or indirectly. This path is referred to as chemical signalling [14].

##### Short-Chain Fatty Acids (SCFAs)

Short-chain fatty acids (SCFAs) are saturated fatty acids with a maximal chain length of six carbon atoms synthesised by colonic bacteria [26]. SCFAs derive from polysaccharides found in dietary fibres, which cannot be broken down by our own digestive enzymes. In contrast, bacteria in the intestines digest those through a process of anaerobic fermentation. The two most important bacteria in this SCFA production are *Bacteroides* spp. and *Clostridiae* spp. [27,28,29]. Acetate, propionate, and butyrate are the most abundant SCFAs in the colon [20,26].

After their synthesis, SCFAs might exert a direct impact on the gut and support local gut health, or they may be distributed by systemic blood circulation throughout the body, affecting other organs, including the brain. In the gut itself, SCFAs are quickly absorbed by colonocytes and act as the main source of energy for the intestinal lining. Absorbed SCFAs influence the cell’s metabolism. They play an important role in increasing mucous production within the GI tract and enhance the integrity of the gut barrier by upregulating the expression of tight junction proteins. Tight junctions are specific connections between cells (here gastrointestinal epithelial cells), contributing to a physical barrier. A functional gut barrier is crucial for preventing the entrance of pathogens and waste products into the body whilst enabling the uptake of important molecules and nutrients. It also prevents the body from systemic inflammation, which is associated with numerous diseases. Prevention of such systemic inflammation further reduces the risk for subsequent neuroinflammation because through systemic inflammation, the blood–brain barrier (BBB) might be impaired. Neuroinflammation, in turn, leads to compromised brain health [18,26,29]. This sums up the immune pathway of SCFA–brain interaction. In addition, tight junctions are an integral component of the BBB. A regulatory effect of SCFAs on the permeability of this barrier was shown in animal models [29].

Systemically, SCFAs can have direct or indirect effects on the MGBA. The indirect pathway is mediated by affecting the immune or the endocrine system, while direct signalling is the result of the neuroactive characteristics of SCFAs. Once SCFAs reach the brain through systemic circulation, they can cross the BBB and bind to specific G-protein-coupled receptors (GPCRs) expressed in CNS tissue. GPCR-activation may then influence the gene expression (epigenetic modulation) in the correspondent nerve cells. For example, the inhibition of histone deacetylation results in more transcriptionally active chromatin (hyperacetylated histones). The inhibited enzymes, histone deacetylases (HDACs), are thought to be associated with brain development and also with a variety of neuropsychiatric disorders such as depression, Alzheimer’s disease, or schizophrenia [18,20,26].

In addition to the two abovementioned examples, other possible mechanisms by which SCFAs affect the brain include their impact on modulating the levels of neurotrophic factors such as brain-derived neurotrophic factor (BDNF), the synthesis of serotonin, and the production of enteroendocrine cell-derived hormones such as GLP1 or PYY [26].

##### Amino Acids and Neurotransmitters

Some gut bacteria possess the ability to produce neuroactive substances, including amino acids and neurotransmitters [14]. As exemplarily shown in Table 1, the inhibitory neurotransmitter GABA can be produced by *Lactobacillus* spp. or *Bifidobacterium* spp. Acetylcholine is synthesised by *Lactobacillus* spp. Noradrenaline is produced by *Bacillus* spp., *Escherichia* spp., and *Saccharomyces* spp. Serotonin is produced by *Streptococcus* spp., *Candida* spp., *Enterococcus* spp., and *Escherichia* spp. Dopamine is produced by *Bacillus* spp. [18,20]. 

Those microbial neurotransmitters are thought to have a more local than peripheral effect. They might cross the intestinal epithelial cells, but their area of action is limited to the surrounding ENS. Even upon reaching the bloodstream, they will be unable to cross the BBB and therefore cannot enter the brain and are unable to influence brain chemistry. Indirect influences on the brain, by acting on the ENS, are possible. For comprehensive reviews, see [14,20].

In contrast, certain precursors of neurotransmitters, namely amino acids, are capable of crossing the BBB [30]. The levels of circulating tryptophan, the precursor for serotonin (5-HT), can be elevated by *Bifidobacterium infantis* [20]. Within the brain, tryptophan can then be further the bolized into 5-HT and impact brain chemistry [20,28,30]. Similarly, the genus *Bifidobacterium* increases the level of phenylalanine, the precursor for the amino acid tyrosine, which itself again is a precursor for the two neurotransmitters dopamine and noradrenaline in the brain [27]. Increased or decreased abundance of such precursor-producing bacteria shifts the availability of monoamine neurotransmitters in the brain and paves the way for related diseases and behavioural changes [30].

#### 3.2.2. Immune System Signalling

Intestines harbour the largest number of immune cells in the human body, allowing the body to scan all the transitioning food and also the symbiotic gut-residing bacteria for potential pathogens or toxins [21]. With the help of pathogen-associated molecular pattern (PAMP) recognition receptors, intestinal cells can detect certain parts of bacteria such as lipopolysaccharide (LPS). LPS is a characteristic cell wall component of Gram-negative bacteria. Innate immune cells are equipped with Toll-like receptors (TLRs), which are the most studied family of PAMP recognition receptors. LPS activates TLRs. Through this activation, Gram-negative bacterial invasion is identified, and an innate immune response is triggered, which involves many signalling molecules, including cytokines and the recruitment of inflammatory cells [21,31]. Immune signalling (production and peripheral secretion of cytokines) is also mediated to the brain via the bloodstream. Recent research has suggested that in the area of the hypothalamus, cytokines can cross the locally more permeable BBB, although they are generally incapable of crossing the BBB. In the hypothalamus, the major physiological stress response system, the hypothalamus–pituitary–adrenal (HPA) axis, has its starting point. Originally from the gut-derived pro-inflammatory cytokines, interleukin (IL)-1 and IL-6 can activate the HPA axis. Subsequently, body cortisol levels rise, and the body enters the stress response mode [20,21]. Multiple neuropsychiatric diseases and pathological conditions have been linked to a dysregulated stress axis and altered immune signalling within the brain, of which depression and autism spectrum disorder are just two examples [14]. Peripheral systemic inflammation, which is driven by circulating immune signalling molecules such as cytokines, is also a major factor in the pathophysiology of many diseases, including brain disorders. Through systemic inflammation, the BBB becomes disrupted and more permeable for bacterial metabolic products. It is not surprising that the condition of a disrupted BBB facilitates the development of different neuropathologies due to the compromised protective barrier against toxic substances [14].

Another important aspect of the intertwining of the gut microbiome and the immune system is found in microglia cells. Microglia cells are the major component of the brain’s immune system and belong to the group of innate immune cells. They register changes in the molecular milieu in the brain. Activated microglia cells trigger an immune response by producing different cytokines and chemokines. Therefore, they are key players in neuroinflammatory processes and in the regulation of brain homeostasis [21]. Their development and maturation highly depend on gut bacteria, as a wide diversity in bacteria strains is important [14,21]. As a result, dysfunction of microglial cells due to reduced complexity in the microbiome could be linked to neurodegenerative or behavioural disorders, where neuroinflammation and disturbed tissue homeostasis act as driving aetiological factors [14].

#### 3.2.3. Neural Signalling

The intestines are innervated by both ENS and the autonomic nervous system (ANS), of which the vagus nerve (10th cranial nerve) is known to be the most direct connection between the CNS and the gut. Approximately 80% of all nerve fibres in the vagus nerve are afferent; the remaining 20% carry efferent signals from the CNS to the periphery. With both efferent and afferent fibres, the basis for bidirectional communication is already given, either bottom-up (gut to the brain via afferent fibres) or top-down (efferent fibres from the brain to the gut) [14,21]. ANS and ENS are both implicated in the physiological homeostasis of the gut and regulate motility, mucous production, and transition time [21]. The initial formation of the ENS takes place during embryogenesis, but differentiation and maturation processes occur later on after birth. These processes are thought to be influenced by gut microbial development, which takes place simultaneously [14,15,21]. ENS and CNS are functionally interconnected through the use of the same signalling molecules [15,21]. Hence, the identified effects that gut microbiota has on local neurons resemble those discussed earlier in the above paragraphs discussing chemical and immune signalling. Those mechanisms include activation of PAMP recognition receptors such as TLRs through LPS, SCFA signalling, and stimulation of mechanoreceptors or chemoreceptors, which sense neurotransmitters, hormones, or metabolic products. Local neuronal stimulation on the ENS level by bacterial metabolites is thought to be then transmitted to the brain via vagal nerve conduction [14]. Metabolites and neurotransmitters can either be directly produced by gut bacteria or secreted by enteroendocrine cells (EECs), which are also affected by gut microbes [14,21].

In addition, the very physical link from the gut to the brain is seemingly important for some neurodegenerative pathogeneses. In Parkinson’s disease, for example, the accumulation of α-synuclein protein within the brain and neurons is the prime disease hallmark. It is suggested that certain gut bacteria promote α-synuclein accumulation in intestinal nerve fibres, which are hypothesised to be then transported along the axons of the vagus nerve to the CNS, where they spread even further from neuron to neuron through trans-synaptic transport [20,32]. This hypothesis is backed by the fact that α-synuclein aggregations were detected in the ENS neurons of the submucosal and myenteric plexuses much earlier than they could be found in the brain [20,32] and highlights the importance of the communication between the ENS, CNS, and the gut microbiome in the pathophysiology of brain disorders [21].

## 4. Changes in Gut Microbiota and Metabolites in Brain-Related Pathologies

### 4.1. Attention Deficit Disorder with Hyperactivity

Attention deficit hyperactivity disorder (ADHD) is a neurodevelopmental disease with an early onset consisting of the three main symptoms inattentiveness, impulsivity, and/or hyperactivity [1,27]. With a worldwide prevalence of 5.29%, ADHD is one of the most common mental illnesses amongst children and adolescents [1,2]. In total, 70–80% of the risk for developing ADHD is made up of genetic components, whereas the remaining 20–30% has been linked to environmental risk factors such as premature birth, toxins, diet, and psychosocial distress [1,18,33,34]. The exact underlying mechanisms of aetiopathogenesis remain unclear [35]. Several known pathomechanisms in ADHD, both direct and indirect, have been linked to the gastrointestinal microbiota by three possible ways of interaction: (i) direct production of neuroactive metabolites/neurotransmitters, (ii) vagal nerve stimulation, (iii) interactions via the immune system [28,33,36].

One widely accepted hypothesis on the ADHD pathophysiology highlights the dysfunction of the monoamine neurotransmitters dopamine (DA), noradrenaline (NE), and serotonin (5-HT), which are involved in the rewarding and motivational processes in the brain. This hypothesis is further supported by the fact that the current medications treating ADHD target the monoaminergic system and increase the concentration of the neurotransmitters in the synaptic cleft by blocking their re-uptake [18,27,35,37]. The production of those neurotransmitters can be influenced by gut bacteria.

*Bifidobacterium* of the phylum Actinobacteria, for example, encodes an enzyme known as arogenate dehydratase (ADT), which is important for the production of phenylalanine an essential amino acid. The latter can pass the BBB and is metabolised into the amino acid tyrosine, which can further be turned into DA and then NE in the brain [27,30]. Aarts et al. compared faeces samples of ADHD patients (*n* = 19) to samples of a healthy control group (*n* = 66). They found that the relative abundance of Actinobacteria was increased in ADHD patients, whereas Firmicutes were slightly reduced compared to the control group. No changes in the phylum Bacteroidetes were observed. Within the phylum, Actinobacterium, especially the genus *Bifidobacterium,* marked a significant increase [30]. As mentioned above, *Bifidobacterium* is equipped with ADT and thus involved in dopamine synthesis, which is an important neurotransmitter within the brain reward response, which is dysfunctional in patients with ADHD. Therefore, these findings suggest that there is a link between the microbiome and ADHD [28,30].

Whereas the work conducted by Aarts et al. displays a potential mode of interaction between intestinal dysbiosis and neurotransmitter disturbance, similar research is needed focusing on the involvement of gut bacteria and precursors of NE or 5-HT (i.e., tyrosine and tryptophan) in the pathophysiology of ADHD, as no such studies could be identified. It is suggested that a similar mode of action applies and that the precursors produced by bacteria are absorbed by the gut epithelium and carried to the brain via blood circulation, where they can cross the BBB and potentially influence the brain chemistry and the monoamine-induced functioning [18,30].

ADHD patients exhibited a lower level of circulating 5-HT in comparison with healthy individuals in a clinical study. A preclinical study with germ-free (GF) mice confirmed these [28,36]. The precursor of 5-HT, amino acid tryptophan, can be synthesised by *Bifidobacterium* spp., *Lactobacilli* spp., *Clostridium sporogenes,* and *Clostridium bartettii* [28], and 5-HT itself can be produced by other bacteria namely *Streptococcus* spp., *Enterococcus* spp., and *Escherichia* spp. [27], of which *Bifidobacteria* [27,30,33], *Lactobacilli* [33], and *Clostridia* [30] were altered in patients with ADHD (see Table 2).

BDNF is a neurotrophic factor important for the survival of neurons as well as neurogenesis. The gut microbes are capable of regulating BDNF levels via the production of SCFAs. In ADHD, an increase in SCFAs through a fibre-rich diet was positively correlated with raising BDNF levels, which ultimately might improve ADHD symptoms [27].

Other studies, which compared faeces of ADHD patients and healthy controls, did not report any clear differences in bacteria strains between the two groups and could not draw a clear correlation between *Bifidobacterium*, higher phenylalanine levels, and subsequently dysfunctional dopamine signalling in the rewarding response [27,30].

Overall, only a few studies on changes in gut microbiota in ADHD patients have been published, and further work needs to be conducted to obtain more data and less divergent results [28,33]. Generally, ADHD is associated with a reduction in microbiota diversity, although the findings were relatively heterogenous [28,33]. Moreover, the genera *Bacteroides* and *Bifidobacterium* were identified as possible biomarkers for ADHD in children and adolescents by two research groups. In particular, a decrease in *Bifidobacterium* has been mentioned as a risk factor [28,30]. The findings of the different studies regarding microbial changes in ADHD patients are listed down below in Table 2 and in Figure 2.

Additionally, the altered microbiome could induce gut barrier permeability and bacterial leakage, which leads to a systemic increase in LPS and pro-inflammatory cytokine levels. As mentioned earlier, those cytokines are potent activators for the HPA axis, creating a state of chronic stress [33]. In ADHD patients, such HPA-hyperactivation is detected by measuring morning cortisol levels. In addition, pro-inflammatory cytokines are also involved in microglial activation and neuroinflammation, which have been linked to the pathogenesis of ADHD [27]. Lastly, the ANS with the vagus nerve showed alterations in ADHD patients. To which extent the vagus nerve is involved in the pathogenesis remains unclear, but studies have shown that patients suffer from an underactive parasympathetic and overactive sympathetic nervous system, although the findings were not consistent throughout all research groups [27,36]. As the vagus nerve is also involved in the rewarding system, which is known to be altered in ADHD patients, the hypothesis seems plausible [27,36].

### 4.2. Autistic Spectrum Disorder

Autism spectrum disorder (ASD) describes a heterogenous group of neurodevelopmental disorders, mainly affecting children [3,4]. The diagnosis is based on clinical observation of behavioural aberrations and diagnostic interviews [4,41]. The key characteristics of ASD are deficits in social interaction and communication, along with repetitive patterns of behaviours and cognitive difficulties [3,4]. ASD prevalence has increased in recent years, and 1–1.5% of the world’s population is affected [4]. Current estimations predict that 1 in 68 up to 1 in 36 children suffer from ASD [41,42]. Interestingly, compared to girls, boys show a four- to five-fold higher prevalence [3,42]. Brain development and reorganisation are altered in ASD patients [4]. Some of the findings are increased brain volume, which could be seen in neuroimages, and changes in brain connectivity compared to neurotypical children [4,42,43]. Aetiology is thought to be an interplay of genetics and environmental factors. Approximately 10–20% of ASD cases are attributed to genetic mechanisms [3,41]. Polluted air, maternal infections, high parental age, pesticides, preterm birth, and certain drug exposures during pregnancy are some of the environmental factors which have been associated with an increased risk for ASD [3,4].

Autistic patients often also suffer from GI symptoms such as diarrhoea, bloating, constipation, and abdominal pain. They appear more often in ASD patients than in healthy children, and constipation was found to be the most common GI symptom among them [44]. Thus, it is not surprising that researchers started investigating the role of the MGBA in ASD pathogenesis. Even more so, because the brain’s development, including synapse formation and growth, timely coincides with the maturation of the gut’s microbiome. Both processes mainly happen within a critical developmental time window until the age of two to three years [44].

Many studies revealed that the microbiome of ASD patients is different from those of healthy individuals. Some changes could be proven repeatedly by independent researchers, but also inconsistent and contradictory results were collected. Possible reasons for the discrepancy could lie in the patient’s origin, diet, lifestyle but also methodological differences [45]. Nevertheless, the data has shown that autistic children harbour a shifted ratio between the two phyla Bacteroidetes and Firmicutes in faecal and biopsy samples. Levels of Firmicutes were mostly increased, and those of Bacteroidetes decreased [3,44,45,46]. Additionally, several researchers measured a higher abundance of *Clostridium* spp. and lower levels of *Bifidobacterium* spp. and *Enterococcus* spp. [3,44]. SCFA-producing bacteria such as *Bacteroides* spp., *Clostridia* spp., and *Desulfovibrio* spp. could be found in higher abundance, which is consistent with higher SCFA levels in blood and stool of ASD patients [3,45]. Phylum Actinobacterium, class Betaproteobacteria, genera *Dialister*, *Faecalibacterium*, *Lactobacillus*, and *Ruminococcus* also marked an increased presence [45,46]. ASD patients had decreased levels of bacteria involved in degenerating and fermenting carbohydrates such as *Coprococcus*, *Prevotella*, and *Veilonella* [43,46]. In contrast, *Suturella* could be found in higher abundance, which is important for the regulation of mucosa metabolism and the integrity of the intestinal epithelium [43]. In the future, screening for such bacterial composition aberrations could be helpful in ASD diagnosis and a potential target for novel therapeutic approaches, including faecal microbiota transplantation (FMT) or beneficial probiotic bacteria supplementation [44]. A more detailed list of the studies’ can be found in Table 3 and a visual representation in Figure 3 and Figure 4.The lack of consensus among the studies indicates that further research is needed to describe microbiome characteristics in ASD patients.

The changed microbiome and bacterial metabolites influence the permeability of the intestinal lining. Increased permeability in the gut is linked to reduced tight junctions in the epithelium. A lower abundance of *Lactobacillus* strains, as seen in ASD patients, could be a possible cause for increased permeability regarding their role in tight junction maintenance [44]. Additionally, the levels of tight junction component zonulin were seen to be altered in children with ASD [3]. Hence, bacterial metabolites can cross the gut barrier more easily. Higher levels of circulating LPS were found in ASD patients [44,45]. Its leakage can stimulate the immune system. Indeed, elevated pro-inflammatory cytokine levels, including IL-1β, IL-4, IL-6, IL-8, IFN-γ, TGF-β, and TNF-α, were measured in autistic patients. This subsequently triggers systemic inflammation and compromises the function of the BBB, eventually leading to neuroinflammation [43,45,47]. Post-mortem studies of ASD patients’ brains confirmed BBB impairment and revealed increased microglial activation as a sign of neuroinflammation [44,45]. In living ASD patients, positron emission tomography (PET) scans were performed and indirectly showed increased microglia activation [43]. Synapse malfunction is regarded as a potential result of CNS inflammation [44]. Thus, gut dysbiosis and associated neuroinflammation in early life stages may interfere with neurodevelopment [46].

Concentrations of free aromatic amino acids phenylalanine, tryptophan, and tyrosine were increased in faeces of ASD individuals [46]. Different bacterial metabolites showed altered patterns in ASD children compared to healthy controls. Two of them are the metabolic by-product of *Clostridia*: 3-(3-hydroxyphenyl)-3-hydroxypropionic acid (HPHPA) (an abnormal phenylalanine metabolite) and para-cresol (p-cresol) (which derives from tyrosine metabolism). Increased HPHPA and p-cresol were consistent with the higher *Clostridium* abundance [3,47]. HPHPA can potentially deplete catecholamines in the brain, such as dopamine, adrenaline, and noradrenaline produced by the adrenal gland. This is believed to contribute to typical ASD symptoms such as hyperactivity and stereotypical behaviour [46,47]. P-cresol has also been associated with repetitive behaviour patterns [3]. It can inhibit the enzyme dopamine-beta-hydroxylase, which converts dopamine into noradrenaline and therefore regulates dopamine metabolism in the brain [44,46]. Furthermore, p-cresol acts as a marker for a leaky gut once it is found in the systemic circulation because the GI tract is the only production site [44]. P-cresol is further associated with processes such as apoptosis, DNA damage, and inflammation [46].

Elevated concentrations of the neurotransmitters serotonin and GABA were registered in autistic children [44]. An abnormal serotonergic system within the brain is involved in ASD pathogenesis, considering the pivotal role of this neurotransmitter in the CNS development throughout the foetal period and early stages of life [45]. Tryptophan is the precursor of serotonin, and in autistic individuals, *Burkholderia* spp. and *Pseudomonas* spp., which belong to the phylum Proteobacteria, are thought to be involved with the reinforced tryptophan pathway and the promotion of disordered neurobehaviour [45]. Abnormal GABA and glutamate levels found in ASD patients might also be connected to microbiota changes as some *Bifidobacterium* spp. and *Lactobacillus* are recognised GABA producers [44,45]. Excess glutamate has neurotoxic properties and may induce neuronal apoptosis [44]. The exact mechanism of GABA and glutamate involvement in ASD pathogenesis remains unclear [46].

Finally, in most studies, SCFAs were found in lower concentrations in ASD children compared to healthy control children [44,46]. Their synthesis depends on anaerobic bacteria such as *Bifidobacterium*, *Prevotella*, and *Ruminococcus*, which were decreased in ASD patients [46]. They have the potential to influence early brain development after crossing the BBB since they are involved in the modulation of dopamine and serotonin production [3,44]. The expression of tryptophan 5-hydroxylase, an enzyme in serotonin synthesis, is influenced by SCFAs, as well as the expression of tyrosine hydroxylase, which is of importance in the biosynthesis of dopamine, adrenaline, and noradrenaline [45]. SCFAs further regulate CNS physiology by influencing microglial maturation, neuronal signalling, and epigenetics [45]. Epigenetic alterations are mainly attributed to butyrate, which modulates HDAC activity. HDAC inhibition might impact ASD genesis by changing gene expression. One of the genes affected by these epigenetic processes is the transcription factor cAMP response element-binding protein (CREB), which is known to be implicated in neurodevelopment and brain function processes [29,46]. Furthermore, apoptosis and inflammation processes have been linked to epigenetic alterations mediated by SCFAs [46]. Therefore, it is plausible that SCFA disturbances originating from intestinal dysbiosis are involved in ASD pathogenesis, even though the mechanisms need to be further investigated [45].

### 4.3. Schizophrenia

Schizophrenia is a psychiatric syndrome with a worldwide prevalence of 1% and is rated among the 10 most common causes of disability on a global level [11,60]. The psychotic symptoms range from delusions and hallucinations to disorganised speech. Other symptoms are a decrease in motivation, a diminishment in expressivity, and grossly disorganised or catatonic behaviour. Schizophrenic patients might also suffer from cognitive deficits such as reduced speed in mental processes or memory problems [11,60]. Usually, schizophrenia is firstly diagnosed in early adulthood (late teens or early twenties) but might be preceded by a prodromal phase with minor changes in behaviour and cognitive function [60]. Because of the early onset and the potential for life-long impairments, schizophrenia is associated with a huge burden for both the health care system and the economy. Furthermore, the personal burden of disease should not be neglected as, in addition to the life-long impairments, the mean life expectancy is reduced by 15 years compared to the general population [11]. Twin and family studies concluded that 80% of the risk for schizophrenia could be explained by genetics. No single disease-causing gene could be identified, so it is thought to be a polygenic disorder [11,60]. The remaining 20% are made up of environmental factors such as birth complications, childhood adversity (emotional or physical abuse, neglect, family dysfunction), urban area upbringing, alcohol, drugs, malnutrition, infections, or stress [60,61].

Over the last three decades, research has started to focus on the role of immune functions in schizophrenia pathogenesis. It has been suggested that schizophrenia is a low-grade chronic inflammatory disease. Neuroinflammation in the CNS and the periphery could be associated with schizophrenia, in which overly activated neuroimmune microglia cells play a crucial role [62]. Schizophrenia patients had altered activation of microglia compared to a healthy control group. Alterations in the microbiome, especially in early life, act as a predisposition to immune dysfunction, which has been associated with the development of schizophrenia [62,63]. As discussed earlier, microglial development partly depends on gut microbial influences, a fact that supports a potential involvement of microbiome-derived cues in the pathophysiology of schizophrenia.

An altered microbiome and dysbiosis, along with a disrupted gut barrier and bacterial translocation (leaky gut), have been linked to schizophrenia. Several research groups detected elevated levels of serum biomarkers for microbial translocation, such as circulating LPS, LPS-binding protein (LBP), and soluble CD14 (sCD14) in patients’ blood samples [61,62,63,64,65]. These factors activate the immune system by binding to TLRs. TLR4, which recognises LPS, has been found in higher abundance in patients with schizophrenia [63], possibly leading to the impairment of BBB, neuroinflammation, microglial activation, and neuronal damage with deleterious consequences for cognitive functions [62]. Altogether, these data emphasise a correlation between schizophrenia and the elevated levels of pro-inflammatory cytokines involved in the immune response [61].

Various studies evaluated key changes in microbiome composition in schizophrenia. Only a limited number of findings overlapped. Even the question of whether the overall diversity in gut microbiota was increased or decreased in schizophrenia patients cannot be answered beyond any doubt since the findings regarding alpha- and beta-diversity were inconsistent [63,64,65,66]. A compilation of the bacteria analyses can be found in Table 4, as well as in Figure 5 and Figure 6.The diverse findings suggest that further studies with bigger sample sizes and eradication of cofounding factors are necessary to identify schizophrenia-related bacteria. Even for the bacteria that have been associated more clearly, their biological mechanism and role in the pathophysiology of schizophrenia remain largely unknown [63,66].

Associations between changes in bacterial metabolites found in schizophrenia patients and the disease development are clearer. The impact of translocation biomarkers and cytokines on the pathology of schizophrenia has already been discussed above. Studying faecal samples revealed that schizophrenia patients show alterations in their gut glutamate metabolism. In the gut, the activity of glutamine oxoglutarate aminotransferase (GOGAT), which is involved in glutamate synthesis, was significantly higher than in healthy controls. This results in elevated levels of glutamate [65,66]. The neurotransmitter dopamine is seemingly important for psychosis symptoms such as delusions and hallucinations. In schizophrenia, overstimulation of dopamine receptors D2 in the striatum leading to these symptoms has been postulated. On a pharmacological level, these D2 receptors are blocked by antipsychotic drugs, reducing psychotic symptoms. Dopamine levels in the brain depend at least partly on gut microbiome metabolism, as discussed earlier [61,63]. Furthermore, changes in tryptophan metabolism could be identified, which are also thought to emerge from bacterial metabolism. In addition to being turned into serotonin, tryptophan can also be metabolised through the kynurenine pathway. In Schizophrenia, higher levels of kynurenine metabolites could be measured [61,63,66]. Within the brain, astrocytes metabolise kynurenine into kynurenic acid, which is an antagonist to acetylcholine and glutamate receptors. Those receptors are involved in brain development, behaviour, and cognition. Altered levels of kynurenine and kynurenic acid have therefore been linked to the pathophysiology of schizophrenia [63]. SCFAs are also thought to be of importance. Research findings suggest that their epigenetic modulation potential via decreasing HDAC activity could be crucial in the development of schizophrenia, keeping in mind that 80% of the risk for schizophrenia is attributed to genetics [61,66].

### 4.4. Alzheimer’s Disease

Alzheimer’s disease (AD) is the single most common cause of dementia, making up approximately 60–70% of all dementia cases [7,8]. As it is strongly associated with ageing, it is thought to become even more prevalent in the future with the general population ageing due to increasing life expectancy. Currently, already more than 44 million people are affected on a global level, and therefore it is a rising global health problem [7,8]. AD firstly presents itself with a progression in cognitive impairment, starting with difficulties in episodic memory and remembering new information, further affecting spatial orientation and executive functions. In later stages of the disease, personality changes may occur, and activities of daily life are affected; dressing, eating, and mobility become impossible. This leads to patients’ dependence and disability with a high need for care. AD further reduces life expectancy, as death occurs about 8–10 years after the initial presentation of the disease [7,8].

Ageing and genetic profile are the non-modifiable risk factors for AD [7]. AD is sporadic in most cases (sAD). Rarely is it inherited in an autosomal-dominant way, where symptoms develop as early as the age of 30 to 50 years. Genetic factors are also involved in sporadic AD, contributing to approximately 70% of the risk of developing sAD. Variations in the apolipoprotein E (APOE) gene are the single biggest genetic risk factor for sAD [8,70]. The remaining 30% of the risk is modifiable and connected to environmental and lifestyle factors, including cardiovascular risk factors (hypertension, physical inactivity, obesity, smoking, diabetes) and reduced cognitive activity [7].

The cognitive impairment in AD is caused by irreversible neuronal death and synaptic loss. The accumulation of insoluble and misfolded amyloid-beta (Aβ) proteins is caused by a different secretase cleavage of the amyloid precursor protein (APP) or by reduced clearance of Aβ, where APOE is involved. Aβ then forms extracellular amyloid plaques and accelerates the hyperphosphorylation of tau protein, which forms intraneuronal neurofibrillary tangles (NFTs). Aβ and NFT activate a neurotoxic pathway leading to a loss of neurons and synapses, in which microglia activation and neuroinflammation are crucial [7,8]. At first, the acute neuroinflammatory response helps with Aβ-clearance, but the continuous microglial activation leads to a neurotoxic pathway [71]. In post-mortem AD brains, Aβ-plaques, NFTs, and brain atrophy can be observed with accentuation in the temporal lobe (including hippocampus) and parietal lobe [8].

Increasing evidence shows that the gut microbiota may be contributing to the pathogenesis of AD. As we age, our microbiome undergoes fundamental changes. In the elderly, a decrease in microbial diversity could be found, and they host more pro-inflammatory bacteria and less anti-inflammatory bacteria (Bacteroidetes, *Bifidobacteria*, *Lactobacillus*). Additionally, SCFA-producing bacteria species were found to be less abundant in the elderly. Decreased SCFA levels are thought to facilitate activation of microglial cells by inducing inflammatory processes, first on the gut leading to a leaky gut, but also on a systemic level, including the brain, as the inflammation signalling molecules spread throughout the body. In fact, elevated levels of circulating pro-inflammatory cytokines were found in AD patients [70,72,73,74,75,76]. This, in turn, also affects the permeability of the BBB [70,71,74,76]. The interrupted BBB facilitates neuroinflammation in the brain and neuronal death, as seen in AD [70,74].

Further, the gut microbiota is a source of amyloid proteins. Many bacteria strains are capable of producing amyloid. *Bacillus subtilis*, *Escherichia coli*, *Mycobacterium tuberculosis*, *Salmonella enterica*, *Salmonella typhimurium*, and *Staphylococcus aureus* are a few examples [29,70,76]. Those bacterial amyloids are thought to influence AD pathogenesis in three different ways. Firstly, they may act as an inducer of inflammation response against Aβ. Even though bacterial amyloids and Aβ differ in their amino acid sequence, they resemble each other in their folding structure. Therefore, they are recognised by the same TLRs [29,74,75,76]. TLR activation triggers the inflammatory immune response, which in turn might lead to further neuronal amyloid production inside the brain and reinforces the process of chronic neuroinflammation [74,76]. Secondly, through the concept of molecular mimicry, microbial amyloids could show a prion-like behaviour and cross-seed to the brain, where they promote the formation and accumulation of pathogenic β-sheet structure in other proteins, which leads to misfolded Aβ, as in AD [70,74,77]. Thirdly, an additional hypothesis suggests that bacterial amyloids might be leaking from the gut and find their way to the brain, where they contribute to the brain amyloid load. However, so far, this translocation has only been observed in animal studies with mice [70,76].

In addition to amyloids, (Gram-negative) bacteria also excrete LPS, which has easy access to the brain in aged people with compromised gut barrier and BBB. Plasma LPS levels of AD patients were identified to be up to three times as high as in healthy controls. These elevated levels may contribute to low-grade chronic inflammation, and once LPS comes into contact with microglia cells in the brain, neuroinflammation is triggered by TLR activation [70,76]. Additionally, in AD patients, the presence of LPS, especially in the hippocampus and neocortex (important centres for memory and executive function), has been demonstrated [76,77].

The Association of gut bacteria to AD is being extensively researched, both in animal models and AD patients. Overall, a decreased bacteria diversity within each patient (alpha diversity) and among AD patients (beta diversity) has been observed in comparison to healthy controls [72,76]. The most overlapping findings were an increase in the phylum *Bacteroidetes* [74,76,78] and a decrease in *Bifidobacteria* and Firmicutes [74,76,77]. In general, decreased numbers of anti-inflammatory bacteria (*Bacillus/Bacteroides fragilis*, *Eubacterium rectale*, *Eubacterium hallii*, and *Faecalibacterium prasunitzii*) were detected in AD patients’ stool samples [74]. The detailed list can be seen in Table 5. The data is further visually represented in Figure 7.

### 4.5. Parkinson Disease

Parkinson’s Disease (PD) is, after Alzheimer’s disease, the most prevailing neurodegenerative disease [82]. There is a strong association with age. Approximately 2–3% of people older than 65 are affected by PD, whereas, in the general population, the prevalence is approximately 0.3% [9]. With the increase in worldwide life expectancy, a dramatic increase in prevalence is expected in the future [10]. PD is the most common cause of clinical parkinsonism syndrome, which consists of brady-/hypokinesia, resting tremor, rigidity, and postural instability [82]. In addition to these motor symptoms, there is also a large variety of non-motor symptoms, which might occur much earlier than the motor manifestations. Some of these non-motor symptoms are hyposmia, REM–sleep behavioural disorder, pain, depression, and obstipation [9,10,82]. Aetiology is multifactorial. PD’s heritability lies at 30%. Environmental and lifestyle factors are the major contributors to the overall PD risk [10]. The modifiable risk-increasing factors include exposure to toxic chemicals such as pesticides and head injuries. Caffeine, exercise, and nicotine, on the other hand, seem to have a protective effect [10]. The pathologic hallmarks of PD are the degeneration of dopaminergic neurons in the substantia nigra located in the midbrain and intracellular protein aggregations of misfolded α-synuclein, also known as Lewy bodies [82]. This results in a dopamine deficiency in substantia nigra, a functional component of the basal ganglia, which is instrumental for motor function [10]. Motor symptoms manifest once 50–70% of all dopaminergic neurons have been lost [10]. The tangible mechanisms by which PD is triggered remain elusive, and the pathogenesis appears complex. Oxidative stress, mitochondrial dysfunction, neuroinflammation, impaired autophagy, and protein aggregation are shown to contribute to PD pathogenesis [10,83,84].

During the past two decades, the GI tract and the microbiome have become a major area of research regarding PD aetiopathogenesis. Gut microbial dysbiosis and its effects on the MGBA contribute to PD pathogenesis and progression on various pathways. Changes in bacteria and metabolite abundance, lost gut integrity, deficiency in BBB, and neuroinflammation are important players [51,84].

According to the most popular hypothesis connecting the gut and PD, the aggregation of misfolded α-synuclein might be initiated in the ENS and further propagated to the CNS via retrograde axonal transport, especially through the vagus nerve. Through this cell-to-cell transmission, α-synuclein could finally reach the pars compacta of the substantia nigra [85]. This hypothesis goes back to Braak and colleagues and is supported by the early neuropathological detection of α-synuclein inclusions in the ENS of PD patients. Such inclusions could be found in neurons of the submucosal and myenteric plexus, which are part of the ENS before they could be detected in the brain [32,51]. These early ENS aggregations could, at least partially, explain why gastrointestinal symptoms often precede the onset of the neurological PD symptoms [29]. Animal models have further confirmed the involvement of the vagus nerve in PD pathogenesis [86]. Additionally, population-based studies have shown that people with truncal vagotomy had a reduced risk after >5 years for PD compared to matched controls [83,84,85,86]. It is postulated that once misfolded α-synuclein is formed, it could act as a template for additional misfolding and lead to protein inclusions that can spread from neuron to neuron (prion-like behaviour) [32,84,86]. Accumulating α-synuclein forms oligomers, which can then build fibrils, which can eventually turn into Lewy bodies [32]. These protein aggregates are potent activators of microglial cells and can therefore provoke inflammatory processes, resulting in neuroinflammation and neurodegeneration [84,87]. Additionally, TLRs are also stimulated by α-synuclein [87]. Braak et al. suggested that unknown environment toxins or microbial pathogens may trigger the initial protein aggregation in the gut [84,85,86]. Recent studies have been focusing on the role of gut microbiota in the initiation of α-synuclein aggregation. In PD patients, an increase in intestinal α-synuclein showed a strong correlation with microbial alterations, which lead to inflammation and loss of gut integrity [85]. Additionally, the bacterial inflammatory metabolite, LPS, facilitates α-synuclein accumulation and aggregations, as well as the formation of fibrils [85]. PD patients exhibit elevated levels of serum LPS and decreased levels of LBP, which helps with the elimination of the endotoxin secreted by Gram-negative bacteria [51,88]. Through TLRs, LPS activates the innate immune system, including microglia cells, and leads to inflammation–locally, systemically, as well as in the ENS and CNS. Eventually, this can result in a loss of dopaminergic neurons [83,85,88].

SCFAs are also relevant for PD pathogenesis. In multiple clinical studies, faecal samples of PD patients generally contained a reduced amount of SCFAs compared to healthy individuals [29,51]. This reduction is consistent with the lower abundance of SCFA-producing bacteria (including *Bacteroides*, Blautia, Coprococcus, Faecalibacterium prausnitzii, Lachnospiraceae, and Roseburia) in PD patients [29,51,84,85]. SCFA deficiencies may lead to intestinal and neuronal inflammation, gut leakage, microglial activation, and also Lewy body formation in the ENS, which are all important factors in PD pathogenesis [83,84,85].

Both intestinal inflammation and increased gut permeability could be observed in PD patients [84]. Specific alterations of gut bacteria in PD patients lead to increased gut permeability accompanied by intestinal inflammation and neuroinflammation [87]. In colonic biopsies, PD patients expressed a higher level of pro-inflammatory cytokines such as IF-γ, IL-1β, IL-6, and TNF-α, as well as glial cell markers such as GFAP and Sox-10. Furthermore, stool samples of PD patients also measured elevated cytokines and chemokines [85]. A marker for the impaired gut barrier, the tight junction-associated cytoplasmatic protein zonulin, was significantly elevated in faecal samples of a case–control study on PD patients [86]. Moreover, the elevated LPS and reduced LBP in patients’ serum suggested increased gut permeability [83]. Endotoxins such as LPS and microbes can enter the gut lumen and circulation and induce inflammation and oxidative stress, which supports the formation of ENS and CNS synucleinopathy after activating enteric neurons and glia cells [10,84,87]. This is supported by reports of increased expression of TLR4, a receptor for LPS, in the colon of PD patients [84]. Systemic inflammation can be observed through increased pro-inflammatory cytokine levels and is tearing down the integrity of the BBB. Subsequently, pro-inflammatory factors gain access to the brain, where they create a molecular environment for neuroinflammation and neuronal death [83,84,85,87]. As a consequence, CNS microglial cellsare overactivated. That seemingly affects the dopaminergic neurons in the substantia nigra pars compacta [83,87]. In fact, in the post-mortem midbrains of PD patients, a higher number of immune T-cells could be found, which are suggested to be involved in neuronal death after recognising certain antigen epitopes on α-synuclein molecules [83].

Due to these different intertwined pathways, the gut and its microbes are considered predisposing factors for PD pathogenesis [85]. The question remains which changes in microbiome composition are in common among PD patients. Emerging literature has shown some general features and trends of PD microbiomes, although the variety in research findings is wide. The inconsistencies across different studies could be explained by differences in methodology, geography, age, diet, and cross-sectional study design [83,84]. In general, the bacterial phenotype is shifted towards a pro-inflammatory phenotype, with a higher abundance of bacteria of the genus *Ralstonia*. Conversely, bacteria with anti-inflammatory properties such as genera *Blautia*, *Coprococcus*, *Faecalibacterium,* and *Roseburia* were found in lower quantities in PD patients’ stool samples [51]. Overall, genus *Faecalibacterium*, genus *Prevotella*, and family *Lachnospiraceae* were decreased in PD patients compared to healthy controls. Generea *Akkermansia*, *Bifidobacteriacea*, and families *Christenalleaceae*, *Lactobacillaceae*, *Ruminococcaceae*, and *Verrucomicrobiaceae* were increased [84,86,88]. The results of the different studies are shown in Table 6, as well as in Figure 8 and Figure 9.

### 4.6. Depression

Major depressive disorder (MDD), or simply (major) depression, is a disease imposing a tremendous health burden with a lifetime risk of 15–18%. Approximately every fifth person gets diagnosed with MDD at least once in life. Around the globe estimated 350 million people are suffering from depression. In a World Health Organisation Report of 2017, major depression was listed as the biggest contributor to worldwide disability and suicide [96]. Women are affected twice as much as men [5,97]. In most people, the first episode of a major depression happens between the age of 20 to 40 years. The key symptoms of MDD are persistent sadness, loss of pleasure (anhedonia) and interest, listlessness, and low energy levels. Additionally, a set of side symptoms such as insomnia or loss of appetite are possible [5]. Genetic predisposition contributes one-third to MDD risk. Environmental influences, including epigenetic mechanisms, account for the remaining two-thirds [97]. Various factors are involved in the pathogenesis of depression. Chronic stress exposure is a well-known cause of MDD. It leads to dysregulations of the HPA-axis with elevated plasma cortisol levels [5,97,98]. Subsequently, the body’s internal homeostasis is disrupted [97]. Moreover, changed CNS neurotransmitter levels are thought to be important contributors to MDD pathology. Synaptic concentrations of monoamines (serotonin (5-HT), dopamine, noradrenaline), and GABA are lower in depressive patients [97,98]. Established pharmacological therapies are also targeting these neurotransmitter imbalances [5,97]. In addition, reduced neuroplasticity (including BDNF depletion) and chronic inflammation processes, with pro-inflammatory cytokine and microglia involvement, are also associated with MDD pathogenesis [97].

Currently, several potential connections between gut microbiota and depression are being discussed. Most human studies have been focusing on the composition of microbiota in depressive patients. Studies of possible mechanistic pathways are relatively scarce. Nevertheless, a few ways of interaction were identified.

In general, the taxonomic changes in bacteria were connected to a pro-inflammatory state, with a reduction in anti-inflammatory bacteria (*Faecalibacterium*, Firmicutes, and *Subdoligranulum*) and an increase in pro-inflammatory ones (*Alistipes,* Bacteroidetes, *Eggerthella*, Flavonifractor, and Gammaproteobacteria). These changes go hand in hand with metabolite alterations. Within the group of anti-inflammatory bacteria, SCFA-producing bacteria were reduced (e.g., *Faecalibacterium and Prevotella*) [99,100,101,102]. Accordingly, faecal samples of MDD patients contained lower levels of total SCFAs compared to healthy controls [29]. SCFAs are known to have various interactions with the host’s physiology. In the pathophysiology of depression, their influence ranges from epigenetic mechanisms via HDAC inhibition to downregulation of pro-inflammatory cytokine production, vagus nerve stimulation, microglia maturation, and BDNF production [97,99]. The neurotrophic factor BDNF, which stimulates neurogenesis, was found to be reduced in patients with MDD [5]. The importance of SCFAs is further supported by different clinical trials, where after the intake of probiotics, including SCFA-producing strains, MDD patients experienced a reduction in depressive symptoms, and healthy individuals reported improved mood and cognition [99].

Altogether, the above-mentioned factors lead to local inflammation, which impairs the gut epithelial integrity. This is followed by an increase in systemic inflammation [100,102]. LPS from Gram-negative bacteria could translocate and, via PAMP-activation, stimulate microglia cells and cytokine production in depressive patients [97]. In animals, such an inflammation-associated MDD model could be confirmed. Intravenous LPS injections in rats promoted depression-like behaviour [99]. Additionally, in patients with MDD, an upregulated genetic pathway for the metabolism of LPS was found [99]. Meta-analyses reported elevated cytokine levels in MDD patients, including IL-1β, IL-6, and TNF-α. The low-grade systemic inflammation can also be seen in elevated levels of C-reactive protein (CRP) [51,97,99,100]. Post-mortem studies of depressive patients’ brains found evidence for increased microglial activation and neuroinflammation [5].

Furthermore, disturbances in gut microbiota are thought to affect the production of neurotransmitters, including glutamate and tryptophan metabolism [100]. Changes in GABA metabolism and signalling have been associated with an increased risk for depression and anxiety. In MMD patients, elevated blood levels of GABA were shown. Additionally, its precursor glutamate was found in higher abundance in depressive patients [100]. A depletion in *Bacteroides* in MDD patients might be a contributing factor to GABA alterations [100]. In the future, more studies looking into the interplay of neurotransmitters and gut bacteria are sought to be performed.

It has become clear that stress disturbs the gut microbiome. The intraindividual α-diversity decreases in individuals frequently exposed to stress. Moreover, the microbiota plays a role in the function of the HPA-axis through bacteria metabolites, and stress responses can be mediated by certain bacteria strains through the vagus nerve, affecting central nervous function [101,102,103].

In addition to a shift towards a pro-inflammatory composition as described above, researchers trying to classify taxonomic changes in depressive patients identified the following trends: On the phylum level, Actinobacteria, Bacteroidetes, and Proteobacteria were enriched, whereas Firmicutes decreased. Generally, MDD patients marked a decrease in *Coprococcus*, *Dialister*, *Escheria*, *Faecalibacterium*, *Prevotellaceae*, *Ruminococcus*, *Sutterellaceae*, and *Veillonellaceae*. An increase was found in *Actinomycetaceae*, *Alistipes*, *Atopium*, *Blautia*, *Clostridium*, *Eggerthella*, and *Paraprevotella*. Interestingly, for *Bifidobacterium,* the literature was especially heterogeneous, and both increases and decreases have been reported [99,100,101,102,103,104]. Although inconsistent findings were made, the α-diversity in MDD patients is assumingly lower [101,102]. More details on the most common bacterial alterations in MDD patients can be found in Table 7 and in Figure 10 and Figure 11.

### 4.7. Bipolar Disorder

Bipolar disorders (BD) are a group of chronic affective disorders and include bipolar disorders I and II. BD I can be diagnosed if recurring manic episodes are present, which may be alternating with depressive episodes. Manic symptoms should be present for a minimum of one week and include reduced need for sleep, general disinhibition, elevated mood, logorrhoea, grandiosity, increased confidence, increased energy, and activity. The depressive episodes fulfil the criteria of a major depressive episode for at least two weeks (depressed mood, loss of interest, anhedonia, fatigue). If the diagnostic threshold for a manic episode is not reached, the episode is classified as hypomania. Recurring hypomanic episodes with or without depressive episodes define BD II [6,12]. Typically, BD starts around the age of 20 years, and it has a lifetime prevalence of 2.4% [6]. BD often leads to disability and impairs patients on a psychosocial level. Overall, BD patients’ life expectancy is reduced by up to 10–20 years due to higher suicide rates and comorbidities such as cardiovascular diseases [6,12]. Twin studies showed a high heritability in BD of approximately 70%, and in genome-wide association studies, various genes could be identified, which are thought to contribute to BD pathogenesis with small effect sizes each [6,12]. The pathogenesis of BD remains unknown. Nevertheless, processes such as inflammation (elevated IL-6 levels in BD patients), increased oxidative and nitrosative stress, epigenetic mechanisms, monoaminergic signalling, disturbance in neuronal and glial plasticity (including BDNF), HPA-dysregulation, and mitochondrial function are associated with the development of BD [6,12,114].

In recent years, possible interactions between BD pathogenesis and the MGBA have been identified. Firstly, some studies observed a reduction in SCFA-producing bacteria in BD patients’ microbiota composition. Especially butyrate-producing bacteria, including *Coproccus*, *Faecalibacterium*, and *Roseburia,* were found in lower abundance. Butyrate is thought to have a crucial impact on brain plasticity because it can stimulate BDNF production in the CNS. Lower BDNF levels, in turn, could contribute to BD development [115,116]. Serum levels of BDNF were lower compared to healthy controls during depressive episodes [114].

Secondly, inflammation in the periphery and the CNS seems to be connected to BD pathogenesis. In BD patients, elevated levels of inflammatory markers such as C-reactive protein (CRP), IL-1, IL-6, and TNF-α were reported [12,117]. The elevation was further accentuated during mood episodes [117,118]. There are multiple factors contributing to this pro-inflammatory state: microglial activation, leaky gut, and HPA-axis activation being three of them [117]. A leaky gut is a result of intestinal inflammation deriving from pro-inflammatory microbiota (e.g., a reduction in *Faecalibacterium* and *Ruminococcaceae*). As already discussed, the combination of a leaky gut and pro-inflammatory bacteria can promote systemic and central inflammation through bacteria migration, LPS increase, TLR-activation, and cytokine release [115,116,117,118]. As a result, the BBB permeability increases and the inflammation can spread to the CNS [115]. Bipolar patients showed higher levels of soluble CD14, which is a marker for translocating bacteria in the context of a leaky gut [117,118]. Furthermore, changes in the tight junction proteins claudin-5 and zonulin in BD patients are an indication of increased intestinal permeability [115]. The inflammatory cytokines IL-1α, IL-1β, IL-6, and TNF-α, were repeatedly measured in increased amounts in BD patients, especially during active mood episodes [115,117,118]. They stimulate the HPA-axis and increase microglial activation in the CNS. The latter effects further contribute to inflammatory processes. Ultimately, the inflammation shows effects on cognition and behaviour, as it is shown that the inflammation mediators may influence neurotransmitter levels (dopamine, noradrenaline, and serotonin) [117].

Microglia cells are involved in synaptic pruning, defined by processes of synapse elimination that occur between early childhood and the onset of puberty. The gut microbiota can interfere in this process. Hence, the function of neuronal circuits might be changed. In bipolar patients, abnormal neuronal connectivity in the limbic and prefrontal cortex could be identified, which originate from dysfunctional pruning processes during development [115].

Analyses of bipolar patients’ microbiota were performed, trying to map characteristic differences in microbial communities. Among the heterogeneous results, some trends could be identified. Most studies agreed on a decrease in α-diversity in comparison to healthy controls [116,118]. BD individuals harboured a decreased amount of *Faecalibacterium*, a Gram-positive bacterium showing anti-inflammatory properties [118]. Researchers also measured an increase in the phyla Actinobacteria and Proteobacteria, the genus *Bacteroides*, as well as a reduction of *Coprococcus* and *Rumicococcaceae* [102,115,116,118]. The results can be found in Table 8, as well as in Figure 12 and Figure 13.

## 5. Discussion

Numerous studies have been performed on individual diseases trying to link the MGBA to the respective pathologies. In this paper, we analysed the characteristic changes in colonic bacteria and their metabolites overlapping between the selected neuropsychiatric disorders or within a certain patient subgroup. The seven diseases included can be split up into the following subgroups: (i) neurodevelopmental disorders (ADHD and ASD), (ii) neurodegenerative disorders (AD and PD), and (iii) psychiatric disorders (schizophrenia), (iv) affective disorders (MDD and BD).

Dysbiosis was a characteristic of all seven diseases discussed. However, research results appear heterogeneous. Quite often, the results of one paper are contradictory to the findings of another. Thus, general statements about pathologic changes in the microbiota are difficult to make. In the future, further studies are needed to clarify the controversial data. Nevertheless, as summarised in Table 2, Table 3, Table 4, Table 5, Table 6, Table 7 and Table 8, some changes seem to be overlapping (see Table 9).

On the phylum level, Actinobacteria are increased in all diseases except AD, where a decrease was reported. Anti-inflammatory phylum Firmicutes is also decreased in most pathologies. For the pro-inflammatory phylum Bacteroidetes, the results were heterogeneous, with a trend towards a general increase. Since many classes of bacteria with contrary characteristics are subordinated to one phylum, unambiguous results on the phylum level are scarce.

On the genus level, the shift towards a pro-inflammatory gut microbiome becomes especially clear. Pro-inflammatory genera *Alistipes*, *Eggerthella*, and *Flavonifractor,* are generally more abundant, whereas anti-inflammatory genera are less present (*Bifidobacterium*, *Coprococcus*, *Eubacterium*, *Faecalibacterium*, *Lactobacillus*, *Prevotella*, *Roseburia*). The anti-inflammatory species *Faecalibacterium prasunitzii* and *Eubacterium rectale* are also reduced [51,70,71,74,76,99,100,101,102,115,117,118].

Another common pattern among the seven diseases is the involvement of SCFAs. Their overall role in (patho)physiology of brain diseases seems to be beneficial, meaning that the relative absence of SCFAs contributes to disease development and progression [21,29]. In our literature review, SCFA alterations were described as a potential driver for the pathogenesis in all seven diseases. Their mode of interaction is broad and ranges from tight junction modulation to anti-inflammatory effects, epigenetic mechanisms (HDAC), regulation of microglia maturation, and influence on BDNF and neurotransmitter production [26,68,122].

The question arises whether some SCFAs have a more pivotal role than others and whether certain bacteria alterations can be associated with those SCFA changes. Despite often generalised statements regarding SCFA levels, a more detailed breakdown could be identified for most diseases (Table 10). Especially for ADHD, BD, and schizophrenia, the available data is sparse, calling for additional future studies to close this information gap.

SCFAs are mainly produced through the fermentation of carbohydrates such as resistant starch and dietary fibres. In much smaller amounts, SCFA production might also be generated from amino acid fermentation [69]. The level and type of SCFA produced highly depend on the daily fibre intake as well as on the bacterial colonisation in the large intestine [68]. For acetate production, the bacteria are not highly specialised, and many bacteria strains, including *Akkermansia muciniphila*, Bacteroides spp., Bacteroidetes, *Bifidobacterium* spp., *Lactobacillus* spp., *Prevotella* spp., *Ruminococcus* spp., and *Streptococcus* spp., are potent acetate producers [29,67,69]. In contrast, butyrate and propionate, for example, can only be produced by a relatively small number of different bacteria [69]. For propionate synthesis, *Akkermansia municiphilla*, Bacteroides spp., Bacteroidetes, Dialister spp., *Firmicutes*, and Veillonella spp., are crucial [67,68,69]. *Anaerostipes*, *Clostridium butyricum*, *Coprococcus*, *Eubacterium*, Eubacterium hallii, Eubacterium rectale, Faecalibacterium prausnitzii, Firmicutes, *Lachnospiraceae*, Roseburia spp., and Ruminococcus bromii are important players in butyrate production [21,29,67,68,69].

Human studies of BD patients revealed lower levels of butyrate, which was consistent with the microbiota analysis of reduced butyrate-producing bacteria (Coproccus, Faecalibacterium, Roseburia) [115,116]. As outlined in Table 9 and Table 10, it is tempting to conclude that in PD, a deficiency in Coprococcus, Faecalibacterium, *Lachnospiraceae*, and Roseburia could be associated with a decrease in butyrate. A reduction of the phylum Firmicutes in MDD might be corresponding to the lower levels of acetate in depressive patients. Reduced levels of propionate in MDD patients might be caused by a decrease in Dialister and Coprococcus. In AD, reduced amounts of Eubacterium rectale and Faecalibacterium prausnitzii could explain lower butyrate levels. Decreased levels of butyrate in ASD could be explained by a reduction in butyrate-producing *Eubacterium*.

Regarding the disorder subgroups, hardly any clear conclusions can be drawn, except that the changes in colon bacteria in MDD and BD are mostly congruent. Since depressive episodes are part of the BD, these matching results could be expected and may point to a common role of gut microbiota in both pathologies. A comparison between the two neurodegenerative diseases, AD and PD, is not quite feasible as the bacteria analysed only partly overlap. The same applies to the neurodevelopmental disorders ADHD and ASD.

When comparing the seven diseases on the level of bacterial metabolites, a general decrease in SCFAs can be seen. Only in ASD and schizophrenia does there seems to be an increase in overall SCFAs. Because of missing data, it is considerably difficult to make a comparison between the disease subgroups. Furthermore, there are certain technical limitations to the data. In different studies, SCFA levels were measured in blood serum or plasma, faeces, or urine, which makes a direct comparison of the findings harder [29]. Moreover, faecal SCFA measurements only reflect the part of SCFAs that has not been absorbed. However, it is the absorbed fraction of SCFAs that holds the biological interaction potential with human physiology [26,69]. Additionally, we have seen the large dependence of SCFA production on certain bacteria strains. Furthermore, several other confounding factors in study design (e.g., age, diet, geography, physical activity, medication) make a direct comparison between the studies a challenging task [14,21].

For other bacterial metabolites such as tyrosine, tryptophan, GABA, or 5-HT, no such consistent patterns as in SCFAs could be identified. Glutamate excess was reported in autism spectrum disorder, schizophrenia, and major depressive disorder [44,45,65,66,100]. Reduced glutamate levels contribute to cognitive impairment in AD [126]. Glutamate is known as the main excitatory neurotransmitter in the CNS and acts as a neuroactive communication molecule within the MGBA with involvements in memory and learning processes, ENS sensitivity, and motility [127]. The exact mechanism, however, by which gut bacteria act upon glutamate levels in humans remains unclear. Corynebacterium as well as Lactobacillus strains (*Lactobacillus paracasei*, Lactobacillus plantarum) have been associated with glutamate production [126,127]. Glutamate produced by gut bacteria might interfere with glutamate signalling in pathologies such as AD [126]. Furthermore, gut bacteria may be involved in glutamate modulation. In ASD, lower abundances of *Campylobacter jejuni* strains have been associated with alterations in glutamate metabolism since *Campylobacter jejuni* might activate glutamate synthesis [126].

Together, the pro-inflammatory bacterial shift and decrease in SCFA may lead to a disruption of the intestinal immune and metabolic homeostasis and local inflammation, which can expand throughout the entire body, eventually reaching the brain. First, reduced levels of SCFAs and dysbiosis loosen the gut barrier, enabling bacteria to translocate. LPS from Gram-negative bacteria gain access to the systemic circulation and activate the immune pathway of the MGBA via TLRs [14,18,21,26,29,31]. LPS and their recognition through the immune system are involved in all seven diseases. Moreover, higher levels of circulating LPS were measured in ASD [44,45], schizophrenia [62], AD [70], and PD [83]. The thereby triggered inflammation response is manifested in all seven diseases with increased levels of pro-inflammatory cytokines, such as IL-1, IL-6, and TNF-α. The primarily local inflammation can spread to a systemic level, which can compromise the BBB integrity. Finally, the same pro-inflammatory molecules can reach the brain and induce neuroinflammation and neuronal death [18,26,29,43,45,47]. Within this central inflammation process, CNS microglia cells are important actors. These residing innate immune cells belong to the macrophage group and may be activated through different mechanisms [128]. Prevalent activating stimuli across the seven diseases of interest are: (i) direct activation through LPS–TLR interaction inside the brain (especially TLR4) [128], (ii) activation through pro-inflammatory cytokines [129], and (iii) facilitated activation and changed maturation due to the absence of SCFAs [68,122]. In AD and PD, resting microglia can be activated through an additional pathological stimulus. The aberrant proteins amyloid-β in AD and α-synuclein in PD are potent inducers of a classical microglial activation [128].

The seven diseases seem to share a final common pathway with excessively active microglial cells [130], chronic neuroinflammation, and the consequent neuronal death. Depending on where in the brain the inflammation happens and which cells are affected, patients may present different symptoms, affecting mood, cognition, and behaviour. Therefore, all the diseases discussed could be summarised as nervous inflammatory diseases, where the MGBA markedly contributes to the disease development and progression [68].

Our review has several limitations. The unavailability of raw data did not allow us to verify the results ourselves. Furthermore, not all the reviewed research papers were of the same quality: Different sample sizes, unbalanced age- and sex-matched sample populations, and methodological heterogeneity of analysis and sample collection made it at times difficult to directly compare the findings of the studies. Age [14,15] and sex [131,132] are known to influence the microbiota population. In particular, the effect of sex hormones on the gut microbiota has been extensively researched in recent years [132]. Although these influences are apparent, some reviewed studies did not fully take them into account by either including only one sex or using unmatched controls. In other studies, the groups were fully matched for sex and age, but in the analysis, gender-driven differences were not further looked into. Therefore, we cannot draw any conclusions on sex-specific changes in relation to the investigated diseases. Other confounding factors which have an influence on the microbiome composition, such as differences in diet and lifestyle behaviour, were not respected in all the studies analysed.

## 6. Conclusions and Future Prospects

In this paper, we attempted to outline overlapping colon microbiota features in patients suffering from different neurodevelopmental, neuropsychiatric, and neurodegenerative diseases (attention deficit hyperactivity disorder, autism spectrum disorder, schizophrenia, Alzheimer’s disease, Parkinson’s disease, major depressive disorder, and bipolar disorder). We have reported that the microbiota–gut–brain axis contributes to the pathophysiology of all these diseases. A consistent feature observed in all seven diseases studied is a shift towards an inflammatory gut microbiome phenotype containing more pro-inflammatory and less anti-inflammatory bacteria strains. The changes on a more detailed level hardly overlap, and results were sometimes contradictory due to confounding factors in the study designs. Future standardised studies will be urgently needed to clarify bacterial changes on lower taxonomic levels and their exact roles in the pathophysiology of these diseases. This will help to establish potential biomarkers for early disease diagnosis. Among the various bacterial metabolites, we highlighted the important role of SCFAs in disease pathogenesis. SCFAs were decreased in most diseases discussed, which is generally associated with a negative effect on the host.

In conclusion, the pathological hallmark underlying the pathophysiology of all discussed diseases is a neuroinflammatory process leading to neuronal cell death. The identified microbiota-gut-brain-axis interactions, including altered SCFA production, are strongly associated with the observed microglia-associated inflammation. We, therefore, propose that these seven diseases can be subsumed under neuroinflammatory diseases.

In the future, the knowledge about microbiome–brain interactions could be used for new treatment approaches. So far, few clinical studies have been performed, experimenting with treatment options ranging from faecal microbiota transplant (FMT) to probiotic supplementation and change of diet. They all have the goal of enhancing useful bacteria strains and reducing the number of harmful bacteria, respectively. Further research is needed to explore the interdependencies between intestinal microbiota and disease pathogenesis to incorporate these novel treatment approaches into clinical practice.

## Figures and Tables

**Figure 1 nutrients-14-02661-f001:**
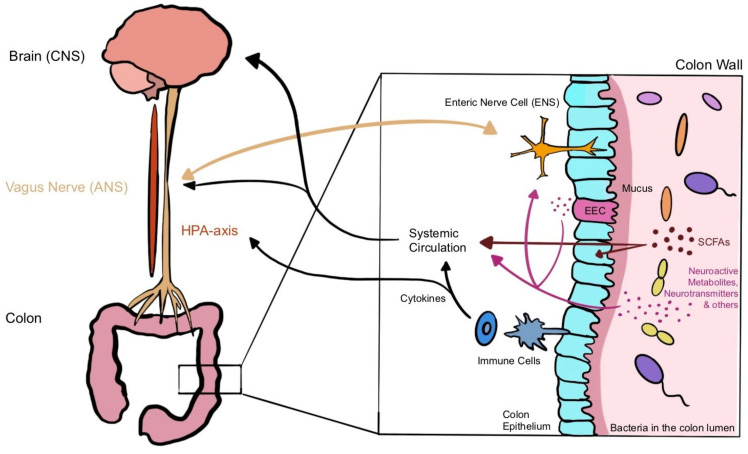
Schematic overview of the bidirectional communication pathways between the gut-microbiota and the brain (Microbiome–Gut–Brain Axis (MGBA)). The colon is inhabited by many different bacteria strains, which directly or indirectly interact with the nerve cells and the brain. The direct pathway includes cytokine signalling after recognition of bacteria through immune cells. Cytokines may also stimulate the body’s stress response via upregulating the hypothalamic–pituitary–adrenal axis (HPA-axis). Indirectly, the bacteria can communicate with their metabolites: utilising dietary fibres, they produce short-chain fatty acids (SCFAs), which are important for epithelium cells and enter the systemic blood circulation. Certain strains of bacteria can produce amino acids or neuroactive metabolites, such as neurotransmitters, which gain access to the brain through blood circulation. Furthermore, bacteria can stimulate enteroendocrine cells, which themselves release hormones into the bloodstream. The bacterial metabolites can also send afferent signals to the brain through enteric nerve cells belonging to the ENS and the vagus nerve. On the other hand, the brain influences important aspects (e.g., motility and mucus secretion) of gut physiology through efferent signals travelling along the vagus nerve and ENS cells. The arrows show the direction of the individual communication pathways and the different signalling stations involved.

**Figure 2 nutrients-14-02661-f002:**
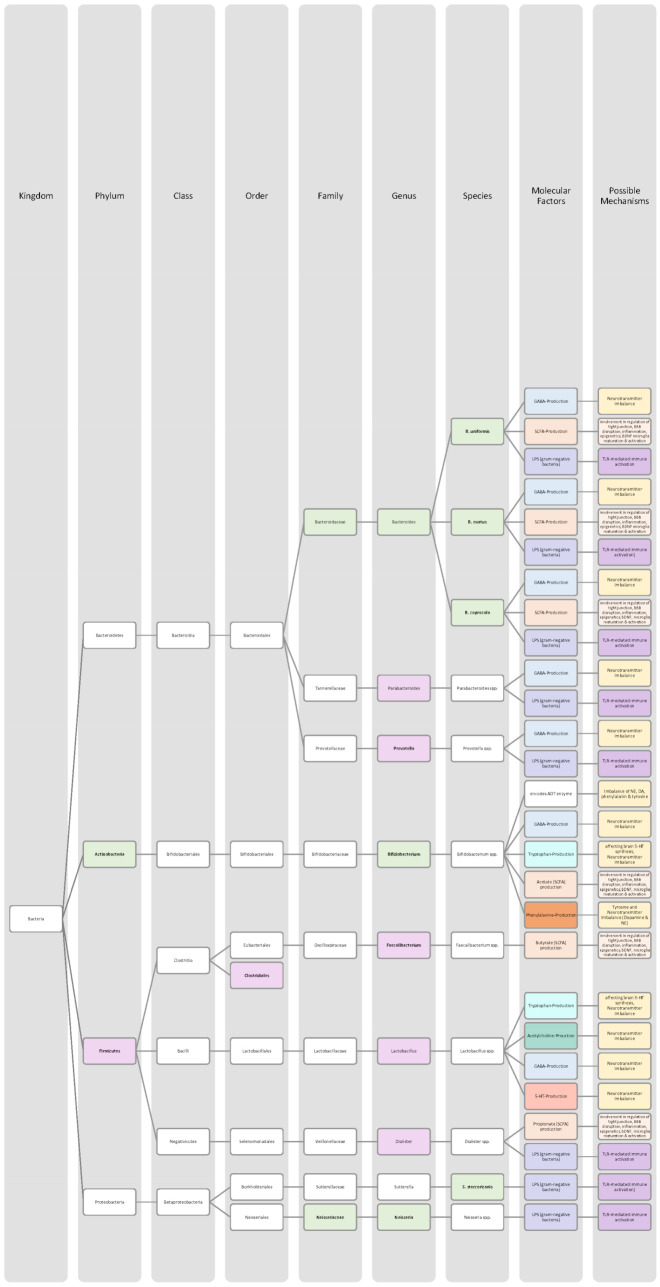
Visual representation of the changes in microbiota composition in ADHD patients from Table 2. The pedigree design highlights the taxonomic classification of the dysbiotic bacteria. Light green cells stand for an increase and the light red cells for a decrease, respectively. Significant changes are shown in bold letters (*p* < 0.05 or LDA values > 2). White cells indicate that no changes were reported. The last two columns contain additional, non-exhaustive information on how the bacteria are thought to impact the pathogenesis. The second last column to the right depicts bacterial molecular factors, which may be components of the bacteria or bacterial metabolites. Finally, the last column lists possible mechanisms by which these molecular factors may interfere with disease development. The colours summarise factors or mechanisms that are repetitive. Bacterial production of GABA (light blue, [36]), tryptophan (turquoise, [28]), acetylcholine (emerald colour, [18,20]), phenylalanine (dark orange, [27]), or 5-HT (salmon colour, [28]) may lead to an imbalance in central nervous system neurotransmitters (yellow). Gram-negative bacteria’s cell wall containing LPS (light purple) can stimulate TLRs located on immune cells (lilac, [21,31]). TLR-mediated immune activation is eventually leading to systemic inflammation, BBB disturbance, and neuroinflammation [14]. SCFAs deriving from bacteria are involved in microglia maturation and activation, BDNF-production, tight junction regulation in the gut barrier as well as in the BBB, epigenetic and inflammatory processes (light orange, [18,20,26,29]). 5-HT = serotonin; ADHD = attention deficit hyperactivity disorder; BBB = blood–brain barrier; BDNF = brain-derived neurotrophic factor; GABA = gamma-Aminobutyric acid; LPS = lipopolysaccharide; SCFA = short-chain fatty acid; TLR = Toll-like receptor.

**Figure 3 nutrients-14-02661-f003:**
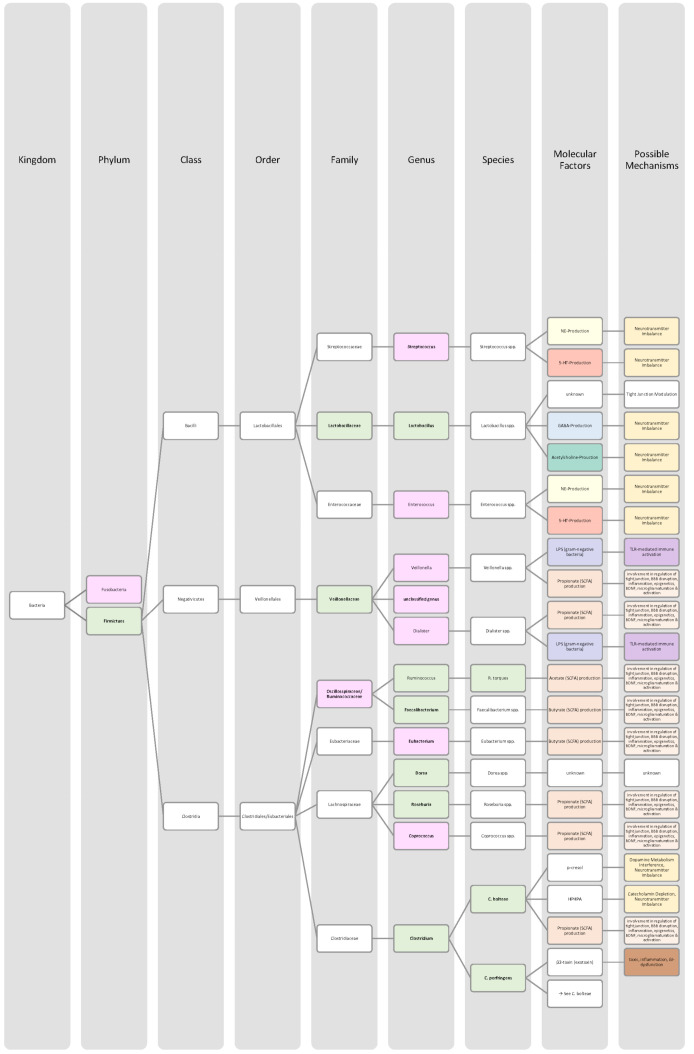
Part 1 of visual representation of the changes in microbiota composition in autism spectrum disorder patients from Table 3. For an explanation, see the legend of Part 2 in Figure 4.

**Figure 4 nutrients-14-02661-f004:**
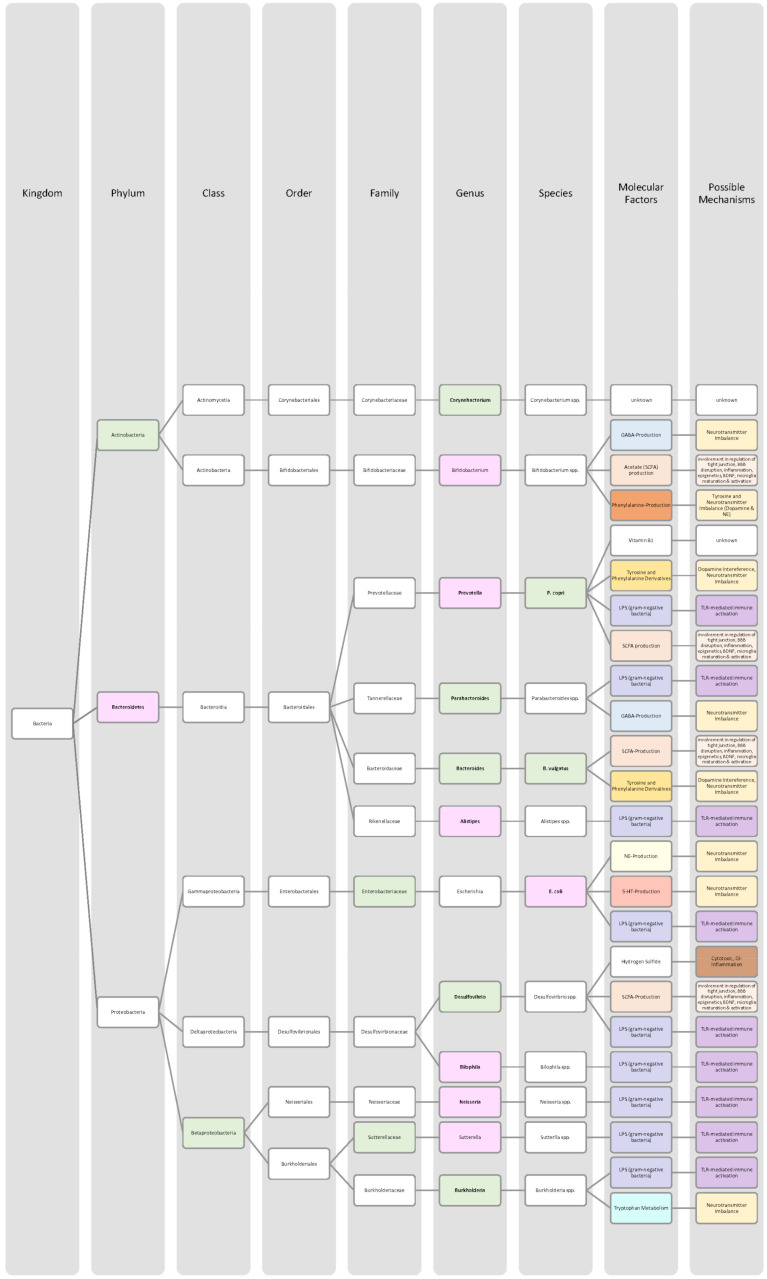
Part 2 of visual representation of the changes in microbiota composition in ASD patients from Table 3. For Figure 3 and Figure 4: The pedigree design highlights the taxonomic classification of the dysbiotic bacteria. Light green cells stand for an increase and the light red cells for a decrease, respectively. Significant changes are shown in bold letters (*p* < 0.05 or LDA values > 2). White cells indicate that no changes were reported. The last two columns contain additional, non-exhaustive information on how the bacteria are thought to impact the pathogenesis. The second last column to the right depicts bacterial molecular factors, which may be components of the bacteria or bacterial metabolites. Finally, the last column lists possible mechanisms by which these molecular factors may interfere with disease development. The colours summarise factors or mechanisms that are repetitive. Bacterial production or involvement in the metabolism of GABA (light blue, [44]), tryptophan (turquoise, [45]), acetylcholine (emerald colour, [18,20]), noradrenaline (light yellow, [18,20,44]), phenylalanine (dark orange, [27]), 5-HT (salmon colour, [18,20,44]) tyrosine and phenylalanine (dark yellow, [45]) may lead to an imbalance in central nervous system neurotransmitters (yellow). Bacterial toxins such as β2-toxin or hydrogen sulfide are cytotoxic for epithelium cells and contribute to GI inflammation (brown) [3,44]. *Clostridium* spp. can further synthesise p-cresol and HPHPA, which both interfere with neurotransmitter balance [3,47]. Prevotella copri produces vitamin B1, but its involvement in pathophysiology remains unknown [44]. Gram-negative bacteria’s cell wall containing LPS (light purple) can stimulate TLRs located on immune cells (lilac, [21,31]). TLR-mediated immune activation is eventually leading to systemic inflammation, BBB disturbance, and neuroinflammation [14]. SCFAs deriving from bacteria are involved in microglia maturation and activation, BDNF-production, tight junction regulation in the gut barrier as well as in the BBB, epigenetic and inflammatory processes (light orange, [18,20,26,29]). If the involvement of the bacterial strain in the disease pathogenesis remains elusive, the last two columns are labelled as unknown, respectively. 5-HT = serotonin; ASD = autism spectrum disorder; BBB = blood–brain barrier; BDNF = brain-derived neurotrophic factor; GABA = gamma-Aminobutyric acid; GI = gastrointestinal; HPHPA = 3-(3-hydroxyphenyl)-3-hydroxypropionic acid; LPS = lipopolysaccharide; NE = noradrenaline; p-cresol = para-cresol; SCFA = short-chain fatty acid; TLR = Toll-like receptor.

**Figure 5 nutrients-14-02661-f005:**
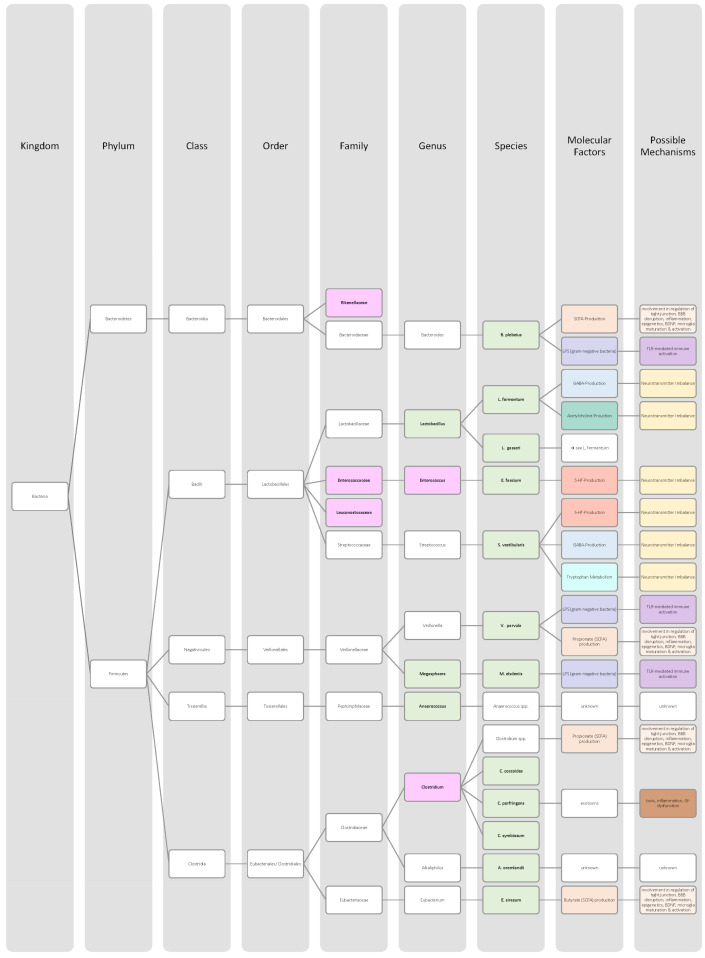
Part 1 of visual representation of the changes in microbiota composition in schizophrenia patients from Table 4. For an explanation, see the legend of Part 2 in Figure 6.

**Figure 6 nutrients-14-02661-f006:**
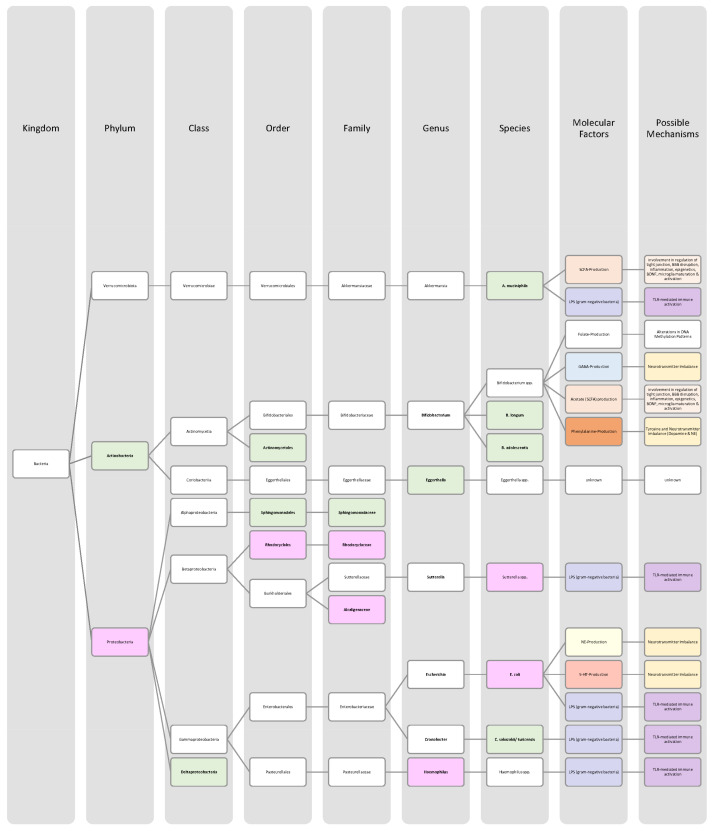
Part 2 of visual representation of the changes in microbiota composition in schizophrenia patients from Table 4. For Figure 5 and Figure 6: The pedigree design highlights the taxonomic classification of the dysbiotic bacteria. Light green cells stand for an increase and the light red cells for a decrease, respectively. Significant changes are shown in bold letters (*p* < 0.05). White cells indicate that no changes were reported or, in the case of Bifidobacterium, both significant increases and decreases were found. The last two columns contain additional, non-exhaustive information on how the bacteria are thought to impact the pathogenesis. The second last column to the right depicts bacterial molecular factors, which may be components of the bacteria or bacterial metabolites. Finally, the last column lists possible mechanisms by which these molecular factors may interfere with disease development. The colours summarise factors or mechanisms that are repetitive. Bacterial production or involvement in the metabolism of GABA (light blue, [18,20,66]), tryptophan (turquoise, [66]), acetylcholine (emerald colour, [18,20]), noradrenaline (light yellow, [18,20]), phenylalanine (dark orange, [27]), and 5-HT (salmon colour, [18,20]) may lead to an imbalance in central nervous system neurotransmitters (yellow). *Clostridium* perfringens synthesises various exotoxins harming the intestinal epithelium lining and causing GI inflammation (brown) [65]. Bifidobacterium spp. produces folate, which contributes to changes in DNA methylation [61]. Gram-negative bacteria’s cell wall containing LPS (light purple) can stimulate TLRs located on immune cells (lilac, [21,31]). TLR-mediated immune activation is eventually leading to systemic inflammation, BBB disturbance, and neuroinflammation [14]. SCFAs deriving from bacteria are involved in microglia maturation and activation, BDNF-production, tight junction regulation in the gut barrier as well as in the BBB, epigenetic and inflammatory processes (light orange, [18,20,26,29,67,68,69]). If the involvement of the bacterial strain in the disease pathogenesis remains elusive, the last two columns are labelled as unknown, respectively. 5-HT = serotonin; BBB = blood–brain barrier; BDNF = brain-derived neurotrophic factor; GABA = gamma-Aminobutyric acid; GI = gastrointestinal; LPS = lipopolysaccharide; NE = noradrenaline; SCFA = short-chain fatty acid; TLR = Toll-like receptor.

**Figure 7 nutrients-14-02661-f007:**
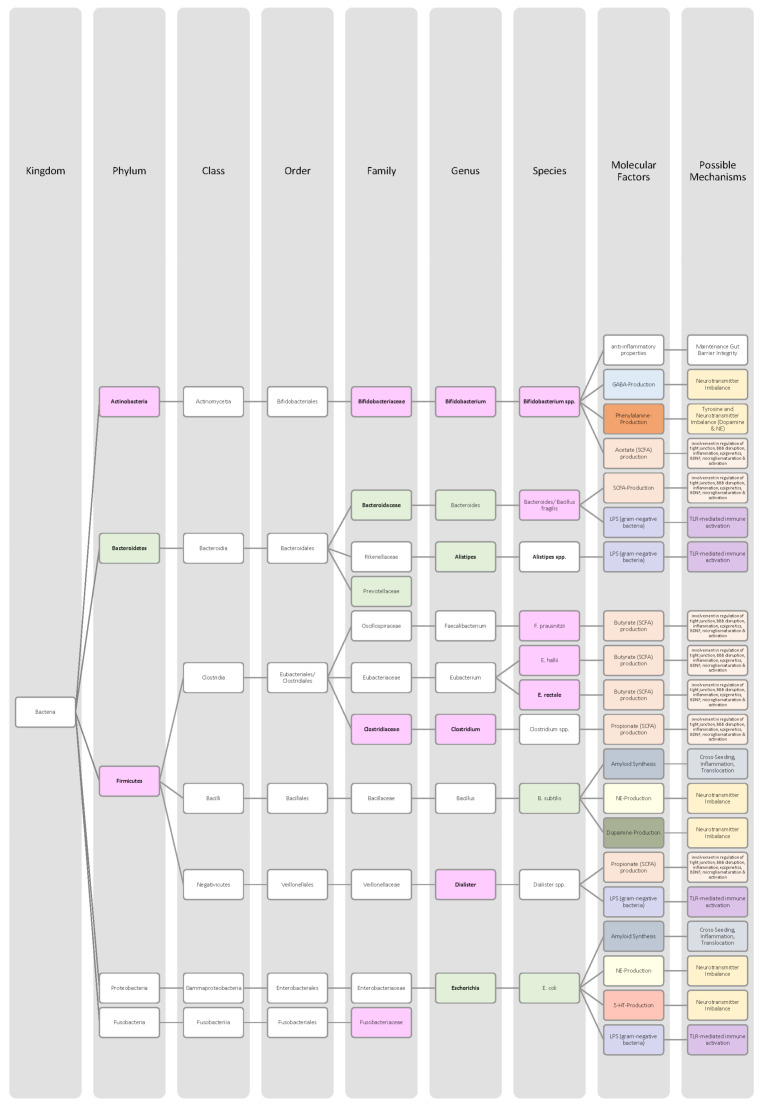
Visual representation of the changes in microbiota composition in Alzheimer’s disease (AD) patients from Table 5. The pedigree design highlights the taxonomic classification of the dysbiotic bacteria. Light green cells stand for an increase and the light red cells for a decrease, respectively. Significant changes are shown in bold letters (*p* < 0.05). White cells indicate that no changes were reported. The last two columns contain additional, non-exhaustive information on how the bacteria are thought to impact the pathogenesis. The second last column to the right depicts bacterial molecular factors, which may be components of the bacteria or bacterial metabolites. Finally, the last column lists possible mechanisms by which these molecular factors may interfere with disease development. The colours summarise factors or mechanisms that are repetitive. Bacterial production or involvement in the metabolism of GABA (light blue, [18,20,70]), noradrenaline (light yellow, [18,20]), dopamine (olive green, [18,20]), phenylalanine (dark orange, [27]), and 5-HT (salmon colour, [18,20]) may lead to an imbalance in central nervous system neurotransmitters (yellow). Bacterial strains such as Escherichia coli and Bacillus subtilis can synthesise bacterial amyloid proteins and are therefore potential contributors to AD amyloid pathology. Bacterial amyloid may induce a (neuro-)inflammatory response, trigger misfolding of neuronal proteins through molecular mimicry and cross-seeding to the brain, or leak from the gut to the brain, where they might accumulate (light and dark blue–grey) [70,74,76,77]. Bifidobacterium spp. maintain the integrity of the gut barrier through their colonisation and anti-inflammatory properties. Thereby they prevent toxins from entering the systemic circulation [70]. Gram-negative bacteria’s cell wall containing LPS (light purple) can stimulate TLRs located on immune cells (lilac, [21,31]). TLR-mediated immune activation is eventually leading to systemic inflammation, BBB disturbance, and neuroinflammation [14]. SCFAs deriving from bacteria are involved in microglia maturation and activation, BDNF-production, tight junction regulation in the gut barrier as well as in the BBB, epigenetic and inflammatory processes (light orange, [21,29,67,68,69]). 5-HT = serotonin; AD = Alzheimer’s disease; BBB = blood–brain barrier; BDNF = brain-derived neurotrophic factor; GABA = gamma-Aminobutyric acid; LPS = lipopolysaccharide; NE = noradrenaline; SCFA = short-chain fatty acid; TLR = Toll-like receptor.

**Figure 8 nutrients-14-02661-f008:**
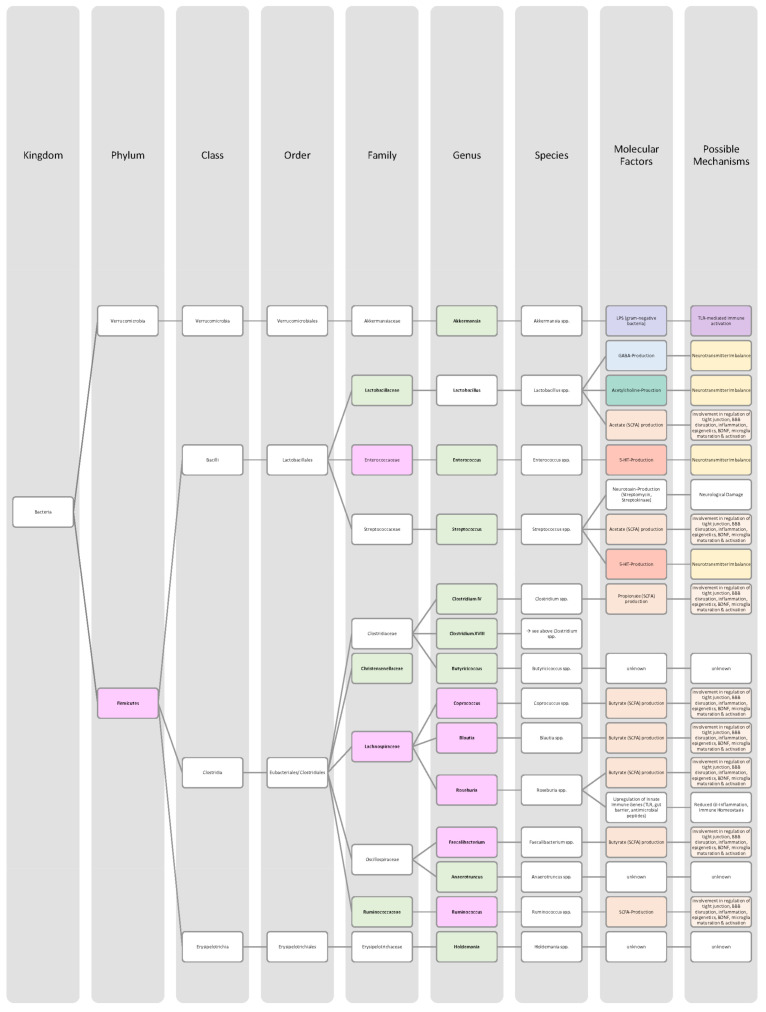
Part 1 of visual representation of the changes in microbiota composition in Parkinson’s disease patients from Table 6. For an explanation, see the legend of Part 2 in Figure 9.

**Figure 9 nutrients-14-02661-f009:**
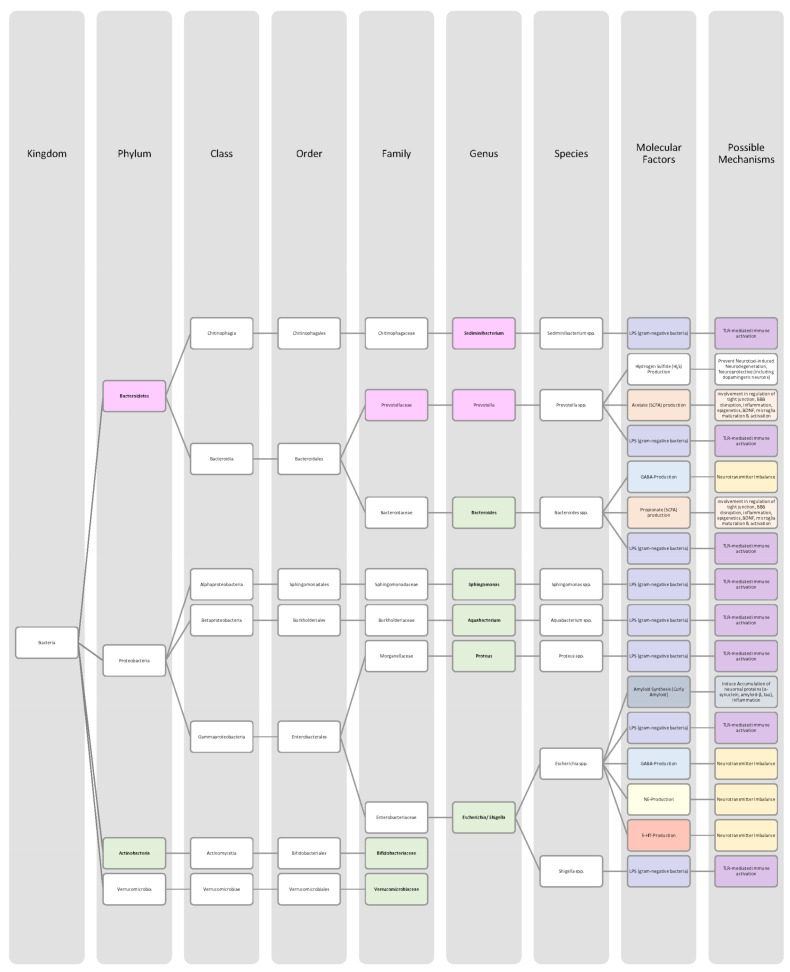
Part 2 of visual representation of the changes in microbiota composition in Parkinson’s disease (PD) patients from Table 6. For Figure 8 and Figure 9: The pedigree design highlights the taxonomic classification of the dysbiotic bacteria. Light green cells stand for an increase and the light red cells for a decrease, respectively. Significant changes are shown in bold letters (*p* < 0.05 or LDA values > 2). White cells indicate that no changes were reported or, in the case of Lactobacillus, both significant increases and decreases were found. The last two columns contain additional, non-exhaustive information on how the bacteria are thought to impact the pathogenesis. The second last column to the right depicts bacterial molecular factors, which may be components of the bacteria or bacterial metabolites. Finally, the last column lists possible mechanisms, by which these molecular factors may interfere with disease development. The colours summarise factors or mechanisms that are repetitive. Bacterial production or involvement in the metabolism of GABA (light blue, [18,20,83,87]), acetylcholine (emerald colour, [18,20]), noradrenaline (light yellow, [18,20]), and 5-HT (salmon colour, [18,20,83]) may lead to an imbalance in central nervous system neurotransmitters (yellow). *Streptococcus* spp. can produce neurotoxins such as streptokinase and streptomycin. These neurotoxins might irreversibly damage neurons, including dopaminergic neurons relevant for PD pathogenesis [51]. Roseburia spp. has anti-inflammatory properties since they upregulate a row of genes involved in the innate immune response (antimicrobial peptides, TLR, intestinal barrier). A reduction in Roseburia spp. thereby contributes to an inflammatory milieu [83]. Prevotella spp. are potent producers of hydrogen sulphide (H_2_S). H_2_S can act as a neuroprotective factor, potentially also for dopaminergic neurons relevant for PD [51,84]. Certain Escherichia spp. might affect the α-synuclein aggregation by synthesising bacterial amyloid proteins (e.g., *E. coli* and curli amyloid). Bacterial amyloid can possibly trigger an immune response and the accumulation of neuronal proteins (α-synuclein, amyloid-β, tau) [83]. Gram-negative bacteria’s cell wall containing LPS (light purple) can stimulate TLRs located on immune cells (lilac, [21,31]). TLR-mediated immune activation is eventually leading to systemic inflammation, BBB disturbance, and neuroinflammation [14,51]. SCFAs deriving from bacteria are involved in microglia maturation and activation, BDNF-production, tight junction regulation in the gut barrier as well as in the BBB, epigenetic and inflammatory processes (light orange, [18,20,26,29,67,68,69,85,86,87,92]). If the involvement of the bacterial strain in the disease pathogenesis remains elusive, the last two columns are labelled as unknown, respectively. 5-HT = serotonin; BBB = blood–brain barrier; BDNF = brain-derived neurotrophic factor; GABA = gamma-Aminobutyric acid; GI = gastrointestinal; H_2_S = hydrogen sulphide; LPS = lipopolysaccharide; NE = noradrenaline; PD = Parkinson’s disease; SCFA = short-chain fatty acid; TLR = Toll-like receptor.

**Figure 10 nutrients-14-02661-f010:**
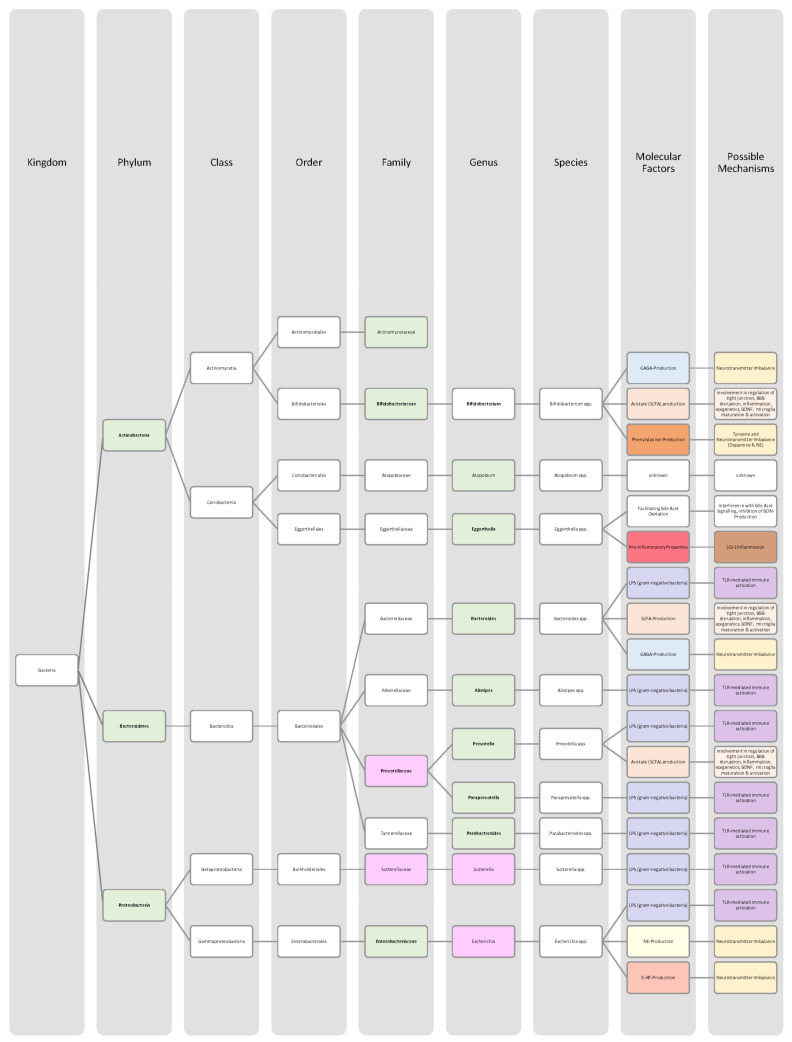
Part 1 of visual representation of the changes in microbiota composition in major depressive disorder (MDD) patients from Table 7. For an explanation, see the legend of Part 2 in Figure 11.

**Figure 11 nutrients-14-02661-f011:**
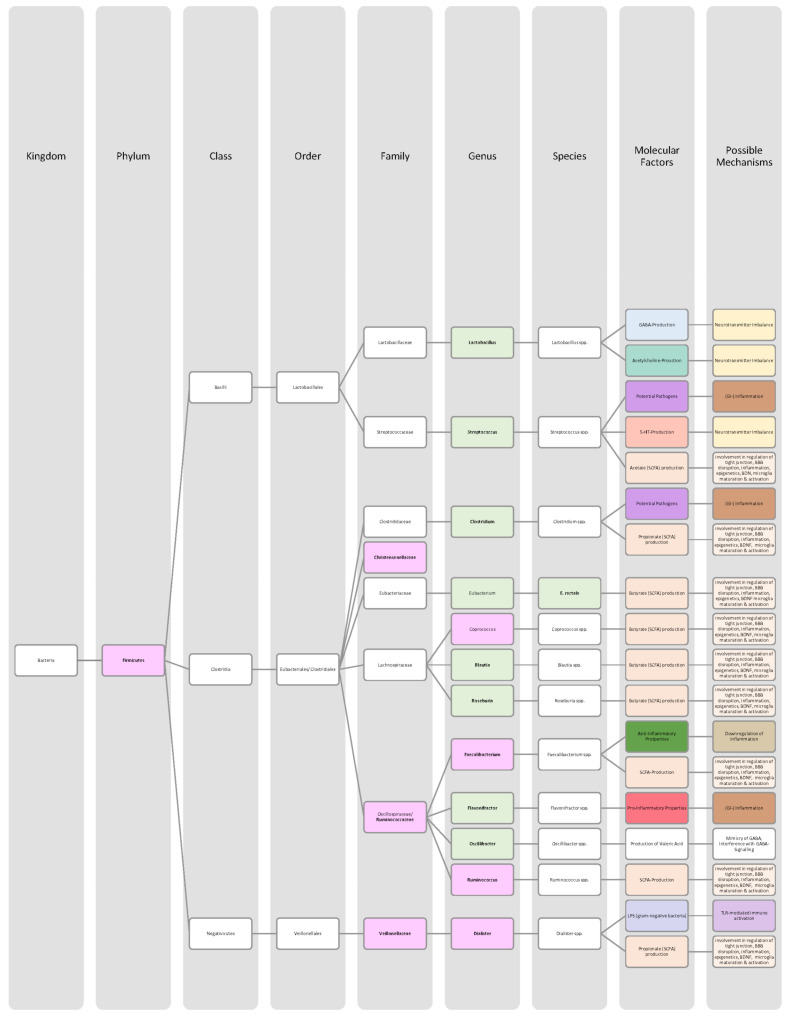
Part 2 of visual representation of the changes in microbiota composition in major depressive disorder (MDD) patients from Table 7. For Figure 10 and Figure 11: The pedigree design highlights the taxonomic classification of the dysbiotic bacteria. Light green cells stand for an increase and the light red cells for a decrease, respectively. Significant changes are shown in bold letters (*p* < 0.05 or LDA values > 2). White cells indicate that no changes were reported or, in the case of Bifidobacterium, both significant increases and decreases were found. The last two columns contain additional, non-exhaustive information on how the bacteria are thought to impact the pathogenesis. The second last column to the right depicts bacterial molecular factors, which may be components of the bacteria or bacterial metabolites. Finally, the last column lists possible mechanisms by which these molecular factors may interfere with disease development. The colours summarise factors or mechanisms that are repetitive. Bacterial production or involvement in the metabolism of GABA (light blue, [18,20,100]), acetylcholine (emerald colour, [18,20]), phenylalanine (dark orange, [27]), noradrenaline (light yellow, [18,20]), and 5-HT (salmon colour, [18,20]) may lead to an imbalance in central nervous system neurotransmitters (yellow). *Eggerthella* spp. and *Flavonifractor* spp. have pro-inflammatory properties (red) and are linked to inflammatory conditions present in MDD (brown) [100]. Additionally, *Eggerthella* spp. are thought to interfere with bile acid signalling pathways because certain strains can facilitate the oxidation of bile acids, which eventually leads to disturbances in BDNF production. BDNF has been found in lower quantities in MDD patients [99]. Some *Streptococcus* spp. and *Clostridium* spp. strains are potential pathogens (purple) and, therefore, contributors to the inflammation component of MDD pathogenesis (brown) [100]. A reduction in *Faecalibacterium* spp., known for their anti-inflammatory properties (dark green), is also associated with inflammatory mechanisms in MDD (beige) [100]. *Oscillibacter* spp. might interfere with GABA signalling through their synthesis of valeric acid, which mimics the molecular structure of GABA and can bind to GABA receptors [100]. Gram-negative bacteria’s cell wall containing LPS (light purple) can stimulate TLRs located on immune cells (lilac, [21,31,99]). TLR-mediated immune activation is eventually leading to systemic inflammation, BBB disturbance, and neuroinflammation [14]. SCFAs deriving from bacteria are involved in microglia maturation and activation, BDNF-production, tight junction regulation in the gut barrier as well as in the BBB, epigenetic and inflammatory processes (light orange, [18,20,26,29,67,68,69,99,100,101,102,104]). If the involvement of the bacterial strain in the disease pathogenesis remains elusive, the last two columns are labelled as unknown, respectively. 5-HT = serotonin; BBB = blood–brain barrier; BDNF = brain-derived neurotrophic factor; GABA = gamma-Aminobutyric acid; GI = gastrointestinal; LPS = lipopolysaccharide; MDD = major depressive disorder; NE = noradrenaline; SCFA = short-chain fatty acid; TLR = Toll-like receptor.

**Figure 12 nutrients-14-02661-f012:**
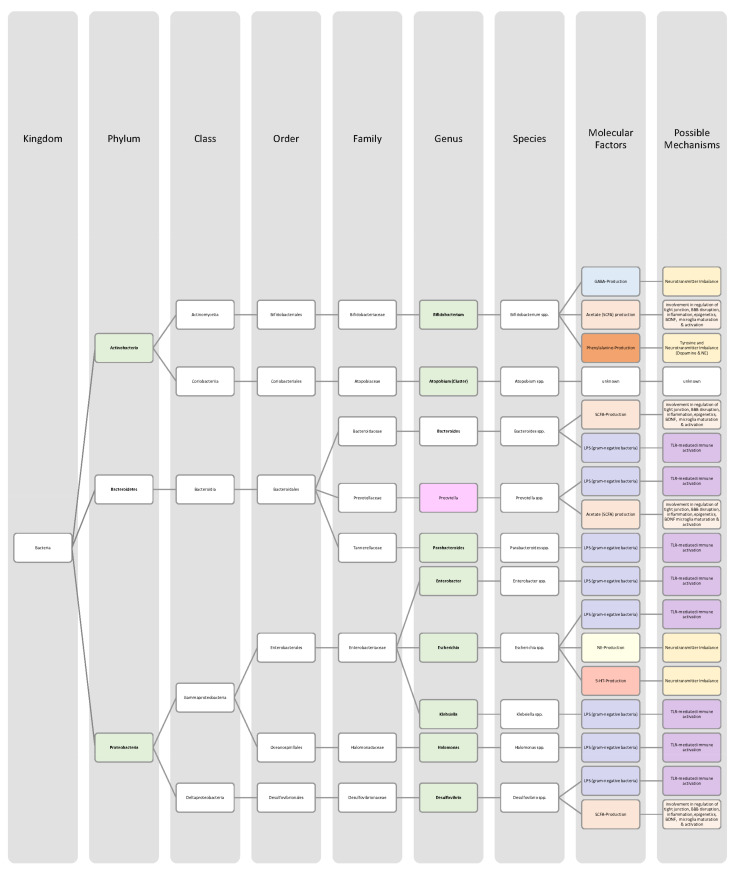
Part 1 of visual representation of the changes in microbiota composition in bipolar disorder (BD) patients from Table 8. For an explanation, see the legend of Part 2 in Figure 13.

**Figure 13 nutrients-14-02661-f013:**
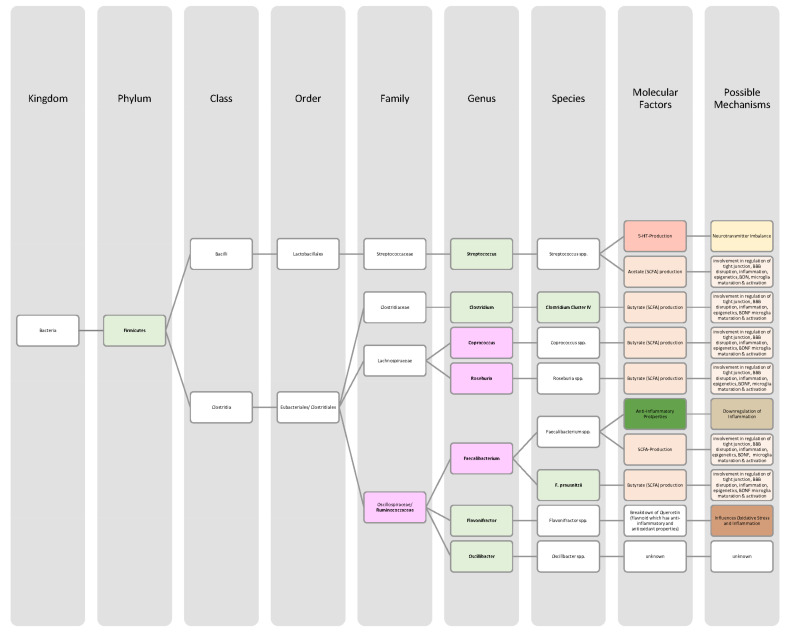
Part 2 of visual representation of the changes in microbiota composition in bipolar disorder (BD) patients from Table 8. For Figure 12 and Figure 13: The pedigree design highlights the taxonomic classification of the dysbiotic bacteria. Light green cells stand for an increase and the light red cells for a decrease, respectively. Significant changes are shown in bold letters (*p* < 0.05 or LDA values > 2). White cells indicate that no changes were reported or, in the case of phylum Bacteroidetes and genus Bacteroides, both significant increases and decreases were found. The last two columns contain additional, non-exhaustive information on how the bacteria are thought to impact the pathogenesis. The second last column to the right depicts bacterial molecular factors, which may be components of the bacteria or bacterial metabolites. Finally, the last column lists possible mechanisms by which these molecular factors may interfere with disease development. The colours summarise factors or mechanisms that are repetitive. Bacterial production or involvement in the metabolism of GABA (light blue, [18,20]), noradrenaline (light yellow, [18,20]), phenylalanine (dark orange, [27]), and 5-HT (salmon colour, [18,20]) may lead to an imbalance in central nervous system neurotransmitters (yellow) [100]. A reduction in *Faecalibacterium* spp., known for their anti-inflammatory properties (dark green), is associated with inflammatory mechanisms in BD (beige) [115,116,118]. *Flavonifractor* spp. can break down the flavonoid quercetin. Flavonoids are polyphenols and secondary metabolites of plants. They are known for their anti-inflammatory and antioxidant effects. The breakdown of quercetin through *Flavonifractor* could contribute to increasing oxidative stress and inflammation [115]. Gram-negative bacteria’s cell wall containing LPS (light purple) can stimulate TLRs located on immune cells (lilac, [21,31]). TLR-mediated immune activation is eventually leading to systemic inflammation, BBB disturbance, and neuroinflammation [14]. SCFAs deriving from bacteria are involved in microglia maturation and activation, BDNF-production, tight junction regulation in the gut barrier as well as in the BBB, epigenetic and inflammatory processes (light orange, [18,20,26,29,67,68,69,115,116]). If the involvement of the bacterial strain in the disease pathogenesis remains elusive, the last two columns are labelled as unknown, respectively. 5-HT = serotonin; BBB = blood–brain barrier; BD = bipolar disorder; BDNF = brain-derived neurotrophic factor; GABA = gamma-Aminobutyric acid; LPS = lipopolysaccharide; NE = noradrenaline; SCFA = short-chain fatty acid; TLR = Toll-like receptor.

**Table 1 nutrients-14-02661-t001:** A non-exhaustive list of major brain neurotransmitters and bacterial strains producing these neurotransmitters.

Neurotransmitter	Bacteria
GABA	*Lactobacillus* spp. [18,20] *Bifidobacterium* spp. [18,20]
Acetylcholine	*Lactobacillus* spp. [18,20]
Noradrenaline	*Bacillus* spp. [18,20] *Escherichia* spp. [18,20] *Saccharomyces* spp. [18,20]
Serotonin	*Streptococcus* spp. [18,20] *Candida* spp. [18,20] *Enterococcus* spp. [18,20] *Escherichia* spp. [18,20]
Dopamine	*Bacillus* spp. [18,20]

**Table 2 nutrients-14-02661-t002:** Bacteria (with taxonomic level) that were found in higher (↑) or lower (↓) abundance in attention deficit hyperactivity disorder (ADHD) patients. Arrows of significant results in at least one study are represented with a grey background. (*p* < 0.05 or LDA values > 2). “-” in the significance column indicate that no statements regarding the significance of the results were made in the corresponding studies. Empty cells mean that no significant differences could be identified. For the studies’ demographic characteristics, sample size and mean age were included. Sample size numbers are split into females and males for patients and healthy controls. The mean age is given for the patient group and the healthy control group, as a further breakdown for the gender groups was not consistently provided. The relevant source is shown in bold numbers.

Bacteria in ADHD Patients	Increase	Decrease	Significance	Sample Size (*n*)	Mean Age (Years)	Source
*Actinobacteria* (*phylum*)	**↑**		*p* = 0.002	96 (♀ 42; ♂ 54) ADHD: 19 Control: 77	ADHD: 19.5 Control: 27,1	[27],[**[30]**]
*Bacteroidaceae* (*family*)	↑		-	31 (♀ 0; ♂ 31) ADHD: 14 Control: 17	ADHD: 11.9 Control: 13.1	[28],[**[38]**]
*Bacteroides* (*genus*)	↑		-	31 (♀ 0; ♂ 31) ADHD: 14 Control: 17	ADHD: 11.9 Control: 13.1	[28],[**[38]**]
*Bacteroides coprocola* (*species*)		**↓**	*p* = 0.028	60 (♀ 19; ♂ 41) ADHD: 30 Control: 30	ADHD: 8.4 Control: 9.3	[28],[**[39]**]
*Bacteroides ovatus* (*species*)	**↑**		*p* = 0.023	60 (♀ 19; ♂ 41) ADHD: 30 Control: 30	ADHD: 8,4 Control: 9.3	[28],[**[39]**]
*Bacteroides uniformis* (*species*)	**↑**		*p* = 0.021	60 (♀ 19; ♂ 41) ADHD: 30 Control: 30	ADHD: 8.4 Control: 9.3	[28],[**[39]**]
*Bacteroidetes* (*phylum*)			*p* = 0.166	96 (♀ 42; ♂ 54) ADHD: 19 Control: 77	ADHD: 19.5 Control: 27,1	[**[30]**]
*Bifidobacterium* (*genus*)	**↑**		*p* = 0.002	96 (♀ 42; ♂ 54) ADHD: 19 Control: 77	ADHD: 19.5 Control: 27.1	[27,28],[**[30]**]
*Clostridiales* (*order*)		**↓**	*p* = 0.003	96 (♀ 42; ♂ 54) ADHD: 19 Control: 77	ADHD: 19.5 Control: 27.1	[**[30]**]
*Dialister* (*genus*)		↓	-	-	-	[**[33]**]
*Faecalibacterium* (*genus*)		**↓**	LDA value > 2	83 (♀ 23; ♂ 60) ADHD: 51 Control: 32	ADHD: 8.47 Control: 8.5	[28,33],[**[40]**]
*Firmicutes* (*phylum*)		**↓**	*p* = 0.001	96 (♀ 42; ♂ 54) ADHD: 19 Control: 77	ADHD: 19.5 Control: 27.1	[**[30]**]
*Lactobacillus* (*genus*)		↓	-	-	-	[**[33]**]
*Neisseria* (*genus*)	**↑**		*p* < 0.05	31 (♀ 0; ♂ 31) ADHD: 14 Control: 17	ADHD: 11.9 Control: 13.1	[28],[**[38]**]
*Neisseriaceae* (*family*)	**↑**		*p* < 0.05	31 (♀ 0; ♂ 31) ADHD: 14 Control: 17	ADHD: 11.9 Control: 13.1	[28],[**[38]**]
*Parabacteroides* (*genus*)		↓	-	-	-	[**[33]**]
*Prevotella* (*genus*)		**↓**	*p* < 0.05	31 (♀ 0; ♂ 31) ADHD: 14 Control: 17	ADHD: 11.9 Control: 13.1	[33],[**[38]**]
*Proteobacteria* (*phylum*)			-	31 (♀ 0; ♂ 31) ADHD: 14 Control: 17	ADHD: 11.9 Control: 13.1	[28],[**[38]**]
*Sutterella stercoricanis* (*species*)	**↑**		*p* = 0.001	60 (♀ 19; ♂ 41) ADHD: 30 Control: 30	ADHD: 8,4 Control: 9.3	[28],[**[39]**]

**Table 3 nutrients-14-02661-t003:** Bacteria (with taxonomic level) were found in higher (↑) or lower (↓) abundance in patients suffering from autistic spectrum disorder (ASD). Included were bacteria strains that were named in at least two different papers. Arrows of the significant results of at least one study are represented with a grey background (*p* < 0.05 or LDA values > 2). *p*-values marked with “*” stem from meta-analyses. “-” in the significance column indicate that no statements regarding the significance of the results were made in the corresponding studies. Empty cells mean that no significant differences could be identified. For the studies’ demographic characteristics, sample size and mean age were included when available. Sample size numbers are split into females and males for patients and healthy controls. The mean age is given for the patient group and the healthy control group, as a further breakdown for the gender groups was not consistently provided. The relevant source is shown in bold numbers.

Bacteria in ASD Patients	Increase	Decrease	Significance	Sample Size (*n*)	Mean Age (Years)	Source
*Actinobacteria* (*phylum*)	↑		*p* = 0.360 *	-	-	[45,47],[**[48]**]
*Alistipes* (*genus*)		**↓**	*p**<* 0.01	80 (♀ 21; ♂ 59) ASD: 40 Control: 40	ASD: 10 Control: 7	[3,44],[**[49]**]
*Alistipes* (*genus*)	↑		*p* = 0.07	30 (♀ 16; ♂ 14) ASD: 10 Control: 10	ASD: 4–10 Control: 4–10	[47],[**[50]**]
*Bacteroides* (*genus*)	**↑**		*p* < 0.001 *	-	-	[45],[**[48]**]
*Bacteroides vulgatus* (*species*)	**↑**		*p* = 0.007	30 (♀ 16; ♂ 14) ASD: 10 Control: 10	ASD: 4–10 Control: 4–10	[3,47],[**[50]**],[51]
*Bacteroidetes* (*phylum*)	**↑**		*p* = 0.001	41 (♀ 12; ♂ 29) ASD: 33 Control: 8	ASD: 2–13 Control: 2–13	[46,51],[**[52]**]
*Bacteroidetes* (*phylum*)		**↓**	*p* = 0.002 *	-	-	[3,43,44,45,46,47],[**[48]**]
*Betaproteobacteria* (*class*)	↑		-	-	-	[44,45,47]
*Bifidobacterium* (*genus*)		**↓**	*p* < 0.001 *	-	-	[3,44,45,46,47],[**[48]**]
*Bilophila* (*genus*)		**↓**	*p* < 0.01	80 (♀ 21; ♂ 59) ASD: 40 Control: 40	ASD: 10 Control: 7	[3,44],[**[49]**]
*Burkholderia* (*genus*)	**↑**		*p* = 0.03	40 (♀ 11; ♂ 29) ASD: 21 Control: 19	ASD: 14.43 Control: 16.05	[3,44],[**[53]**]
*Clostridium* (*genus*)	**↑**		*p* < 0.001 *	-	-	[3,43,44,45,46,47],[**[48]**],[51]
*Clostridium bolteae* (*species*)	**↑**		*p* = 0.01	23 (♀ -; ♂ -) ASD: 15 Control: 8	ASD: - Control: -	[3,47],[**[54]**]
*Clostridium perfringens* (*species*)	**↑**		*p* = 0.031	46 (♀ -; ♂ -) ASD: 33 Control: 13	ASD: 2–9 Control: 2–9	[44,45],[**[55]**]
*Coprococcus* (*genus*)		**↓**	*p* < 0.001 *	-	-	[3,43,44,47],[**[48]**]
*Corynebacterium* (*genus*)	↑		*p* < 0.01	80 (♀ 21; ♂ 59) ASD: 40 Control: 40	ASD: 10 Control: 7	[3,44],[**[49]**]
*Desulfovibrio* (*genus*)	**↑**		*p* = 0.011	41 (♀ 12; ♂ 29) ASD: 33 Control: 8	ASD: 2–13 Control: 2–13	[3,43,44,45,47,51],[**[52]**]
*Dialister* (*genus*)		↓	*p* = 0.760 *	-	-	[3,44],[**[48]**]
*Dialister* (*genus*)	↑		-	-	-	[45]
*Dorea* (*genus*)	**↑**		*p* < 0.01	80 (♀ 21; ♂ 59) ASD: 40 Control: 40	ASD: 10 Control: 7	[3,44],[**[49]**]
*Enterobacteriaceae* (*family*)	↑		*p* = 0.21	54 (♀ 11; ♂ 43) ASD: 30 Control: 24	ASD: 9.5 Control: 9.5	[44,47],[**[56]**]
*Enterococcus* (*genus*)		↓	-	30 (♀ 16; ♂ 14) ASD: 10 Control: 10	ASD: 4–10 Control: 4–10	[44,45,47],[**[50]**]
*Escherichia coli* (*species*)		**↓**	*p* = 0.03	30 (♀ 16; ♂ 14) ASD: 10 Control: 10	ASD: 4–10 Control: 4–10	[44],[**[50]**]
*Eubacterium* (*genus*)		**↓**	LDA > 2.0	50 (♀ 9; ♂ 41) ASD: 30 Control: 20	ASD: 4.43 Control: 4.28	[45],[**[57]**]
*Faecalibacterium* (*genus*)	**↑**		*p* < 0.001 *	-	-	[44,45],[**[48]**]
*Firmicutes* (*phylum*)		**↓**	*p* = 0.001	41 (♀ 12; ♂ 29) ASD: 33 Control: 8	ASD: 2–13 Control: 2–13	[44,46],[**[52]**]
*Firmicutes* (*phylum*)	**↑**		*p* < 0.001 *	-	-	[3,43,44,45,47],[**[48]**]
*Fusobacteria* (*phylum*)		↓	*p* = 0.430 *	-	-	[44],[**[48]**]
*Lachnospiraceae* (*family*)		↓	*p* = 0.1023	50 (♀ 9; ♂ 41) ASD: 30 Control: 20	ASD: 4.43 Control: 4.28	[45],[**[57]**]
*Lachnospiraceae* (*family*)	↑		-	-	-	[44]
*Lactobacillaceae* (*family*)	**↑**		*p* = 0.018	54 (♀ 11; ♂ 43) ASD: 30 Control: 24	ASD: 9.5 Control: 9.5	[51],[**[56]**]
*Lactobacillus* (*genus*)		↓	-	30 (♀ 16; ♂ 14) ASD: 10 Control: 10	ASD: 4–10 Control: 4–10	[44,47],[**[50]**]
*Lactobacillus* (*genus*)	**↑**		*p* < 0.01	80 (♀ 21; ♂ 59) ASD: 40 Control: 40	ASD: 10 Control: 7	[3,43,44,45],[**[49]**],[51]
*Neisseria* (*genus*)		**↓**	*p* = 0.01	40 (♀ 11; ♂ 29) ASD: 21 Control: 19	ASD: 14.43 Control: 16.05	[3,44],[**[53]**]
*Parabacteroides* (*genus*)	**↑**		*p* < 0.001 *	-	-	[44],[**[48]**]
*Parabacteroides* (*genus*)		**↓**	*p* <0.01	80 (♀ 21; ♂ 59) ASD: 40 Control: 40	ASD: 10 Control: 7	[3,44],[**[49]**]
*Prevotella* (*genus*)		**↓**	*p* < 0.05	40 (♀ 5; ♂ 35) ASD: 20 Control: 20	ASD: 6.7 Control: 8.3	[3,43,44,46,47],[**[58]**]
*Prevotella copri* (*species*)	**↑**		*p* = 0.04	30 (♀ 16; ♂ 14) ASD: 10 Control: 10	ASD: 4–10 Control: 4–10	[44],[**[50]**]
*Roseburia* (*genus*)	**↑**		*p* = 0.003	30 (♀ 16; ♂ 14) ASD: 10 Control: 10	ASD: 4–10 Control: 4–10	[44],[**[50]**]
*Ruminococcaceae* (*family*)		**↓**	*p* < 0.001	50 (♀ 9; ♂ 41) ASD: 30 Control: 20	ASD: 4.43 Control: 4.28	[45],[**[57]**]
*Ruminococcus* (*genus*)	↑		*p* = 0.170 *	-	-	[45],[**[48]**]
*Ruminococcus torques* (*species*)	↑		*p* = 0.08	54 (♀ -; ♂ -) ASD: 23 Control: 9	ASD: - Control: -	[3],[**[59]**]
*Streptococcus* (*genus*)		**↓**	*p* = 0.04	30 (♀ 16; ♂ 14) ASD: 10 Control: 10	ASD: 4–10 Control: 4–10	[44],[**[50]**]
*Sutterella* (*genus*)		↓	*p* = 0.480 *	-	-	[44],[**[48]**]
*Sutterella* (*genus*)	**↑**		*p* = 0.05	54 (♀ -; ♂ -) ASD: 23 Control: 9	ASD: - Control: -	[3,43,44],[**[59]**]
*Sutterellaceae* (*family*)	↑		-	30 (♀ 16; ♂ 14) ASD: 10 Control: 10	ASD: 4–10 Control: 4–10	[47],[**[50]**]
*Veillonella* (*genus*)		↓	*p* = 0.460 *	-	-	[3,44],[**[48]**]
*Veillonellaceae* (*family*)	**↑**		*p* = 0.008	54 (♀ 11; ♂ 43) ASD: 30 Control: 24	ASD: 9.5 Control: 9.5	[51],[**[56]**]
*Veillonellaceae* (*unclassified genus of this family*)		**↓**	*p* = 0.04	40 (♀ 5; ♂ 35) ASD: 20 Control: 20	ASD: 6.7 Control: 8.3	[3,43,44],[**[58]**]

**Table 4 nutrients-14-02661-t004:** Bacteria (with taxonomic level) were found in higher (↑) or lower (↓) abundance in patients suffering from schizophrenia. Arrows of the significant results in at least one study are represented with a grey background (*p* < 0.05). “-” in the significance column indicate that no statements regarding the significance of the results were made in the corresponding studies. Empty cells mean that no significant differences could be identified. For the studies’ demographic characteristics, sample size and mean age were included when available. Sample size numbers are split into females and males for patients and healthy controls. The mean age is given for the patient group and the healthy control group, as a further breakdown for the gender groups was not consistently provided. The relevant source is shown in bold numbers. SCZ = schizophrenia.

Bacteria in Schizophrenia Patients	Increase	Decrease	Significance	Sample Size (*n*)	Mean Age (Years)	Source
*Actinobacteria* (*phylum*)	**↑**		*p* = 0.0478	168 (♀ 72; ♂ 89) SCZ: 84 Control: 84	SCZ: 35 Control: 35	[**[65]**]
*Actinomycetales* (*order*)	**↑**		*p* = 0.0025	168 (♀ 72; ♂ 89) SCZ: 84 Control: 84	SCZ: 35 Control: 35	[**[65]**]
*Akkermansia muciniphila* (*species*)	**↑**		*p* < 0.001	168 (♀ 72; ♂ 89) SCZ: 84 Control: 84	SCZ: 35 Control: 35	[**[65]**],[66]
*Alcaligenaceae* (*family*)		**↓**	*p* < 0.001	168 (♀ 72; ♂ 89) SCZ: 84 Control: 84	SCZ: 35 Control: 35	[**[65]**]
*Alkaliphilus oremlandii* (*species*)	**↑**		*p* = 0.008	171 (♀ 84; ♂ 87) SCZ: 90 Control: 81	SCZ: 28.6 Control: 32.8	[**[66]**]
*Anaerococcus* (*genus*)	**↑**		*p* = 0.007	50 (♀ 21; ♂ 29) SCZ: 25 Control: 25	SCZ: 52.9 Control: 54.7	[**[64]**]
*Bacteroides plebeius* (*species*)	**↑**		*p* = 0.0038	171 (♀ 84; ♂ 87) SCZ: 90 Control: 81	SCZ: 28.6 Control: 32.8	[**[66]**]
*Bifidobacterium adolescentis* (*species*)	**↑**		*p* = 0.003	168 (♀ 72; ♂ 89) SCZ: 84 Control: 84	SCZ: 35 Control: 35	[**[65]**],[66]
*Bifidobacterium longum* (*species*)	**↑**		*p* = 0.0075	171 (♀ 84; ♂ 87) SCZ: 90 Control: 81	SCZ: 28.6 Control: 32.8	[**[66]**]
*Bifidobacterium* (*genus*)	**↑**		*p* = 0.0062	171 (♀ 84; ♂ 87) SCZ: 90 Control: 81	SCZ: 28.6 Control: 32.8	[**[66]**]
*Bifidobacterium* (*genus*)		**↓**	*p* = 0.006	171 (♀ 84; ♂ 87) SCZ: 90 Control: 81	SCZ: 28.6 Control: 32.8	[62],[**[66]**]
*Clostridium* (*genus*)		**↓**	*p* = 0.0002	50 (♀ 21; ♂ 29) SCZ: 25 Control: 25	SCZ: 52.9 Control: 54.7	[**[64]**]
*Clostridium *coccoides** (*species*)	**↑**		*p* < 0.001	-	-	[**[62]**]
*Clostridium perfringens* (*species*)	**↑**		*p* < 0.001	168 (♀ 72; ♂ 89) SCZ: 84 Control: 84	SCZ: 35 Control: 35	[**[65]**]
*Clostridium symbiosum* (*species*)	**↑**		*p* = 0.0166	171 (♀ 84; ♂ 87) SCZ: 90 Control: 81	SCZ: 28.6 Control: 32.8	[**[66]**]
*Cronobacter sakazakii/turicensis* (*species*)	**↑**		*p* = 0.0387	171 (♀ 84; ♂ 87) SCZ: 90 Control: 81	SCZ: 28.6 Control: 32.8	[**[66]**]
*Deltaproteobacteria* (*class*)	**↑**		*p* = 0.002	168 (♀ 72; ♂ 89) SCZ: 84 Control: 84	SCZ: 35 Control: 35	[**[65]**]
*Eggerthella* (*genus*)	**↑**		*p* = 0.00307	168 (♀ 72; ♂ 89) SCZ: 84 Control: 84	SCZ: 35 Control: 35	[**[65]**]
*Enterococcaceae* (*family*)		**↓**	*p* < 0.001	168 (♀ 72; ♂ 89) SCZ: 84 Control: 84	SCZ: 35 Control: 35	[**[65]**]
*Enterococcus* (*genus*)		**↓**	*p* < 0.001	168 (♀ 72; ♂ 89) SCZ: 84 Control: 84	SCZ: 35 Control: 35	[**[65]**]
*Enterococcus faecium* (*species*)	**↑**		*p* = 0.0035	171 (♀ 84; ♂ 87) SCZ: 90 Control: 81	SCZ: 28.6 Control: 32.8	[**[66]**]
*Escherichia coli* (*species*)		**↓**	*p* < 0.001	-	-	[**[62]**],[66]
*Eubacterium siraeum* (*species*)	**↑**		*p* = 0.0008	171 (♀ 84; ♂ 87) SCZ: 90 Control: 81	SCZ: 28.6 Control: 32.8	[**[66]**]
*Haemophilus* (*genus*)		**↓**	*p* = 0.004	50 (♀ 21; ♂ 29) SCZ: 25 Control: 25	SCZ: 52.9 Control: 54.7	[**[64]**]
*Lactobacillus fermentum* (*species*)	**↑**		*p* = 0.0026	171 (♀ 84; ♂ 87) SCZ: 90 Control: 81	SCZ: 28.6 Control: 32.8	[**[66]**]
*Lactobacillus gasseri* (*species*)	**↑**		*p* < 0.001	168 (♀ 72; ♂ 89) SCZ: 84 Control: 84	SCZ: 35 Control: 35	[**[65]**]
*Lactobacillus* (*genus*)	**↑**		*p* = 0.027	171 (♀ 84; ♂ 87) SCZ: 90 Control: 81	SCZ: 28.6 Control: 32.8	[62],[**[66]**]
*Leuconostocaceae* (*family*)		**↓**	*p* < 0.001	168 (♀ 72; ♂ 89) SCZ: 84 Control: 84	SCZ: 35 Control: 35	[**[65]**]
*Megasphaera* (*genus*)	**↑**		*p* < 0.001	168 (♀ 72; ♂ 89) SCZ: 84 Control: 84	SCZ: 35 Control: 35	[**[65]**]
*Megasphaera elsdeniis* (*species*)	**↑**		*p* < 0.001	168 (♀ 72; ♂ 89) SCZ: 84 Control: 84	SCZ: 35 Control: 35	[**[65]**]
*Proteobacteria* (*phylum*)		↓	-	50 (♀ 21; ♂ 29) SCZ: 25 Control: 25	SCZ: 52.9 Control: 54.7	[**[64]**]
*Rhodocyclaceae* (*family*)		**↓**	*p* < 0.001	168 (♀ 72; ♂ 89) SCZ: 84 Control: 84	SCZ: 35 Control: 35	[**[65]**]
*Rhodocyclales* (*order*)		**↓**	*p* < 0.001	168 (♀ 72; ♂ 89) SCZ: 84 Control: 84	SCZ: 35 Control: 35	[**[65]**]
*Rikenellaceae* (*family*)		**↓**	*p* = 0.011	168 (♀ 72; ♂ 89) SCZ: 84 Control: 84	SCZ: 35 Control: 35	[**[65]**]
*Streptococcus vestibularis* (*species*)	**↑**		*p* = 0.0036	171 (♀ 84; ♂ 87) SCZ: 90 Control: 81	SCZ: 28.6 Control: 32.8	[**[66]**]
*Sphingomonadaceae* (*family*)	**↑**		*p* < 0.001	168 (♀ 72; ♂ 89) SCZ: 84 Control: 84	SCZ: 35 Control: 35	[**[65]**]
*Sphingomonadales* (*oder*)	**↑**		*p* < 0.001	168 (♀ 72; ♂ 89) SCZ: 84 Control: 84	SCZ: 35 Control: 35	[**[65]**]
*Sutterella* (*genus*)		**↓**	*p* = 0.004	50 (♀ 21; ♂ 29) SCZ: 25 Control: 25	SCZ: 52.9 Control: 54.7	[**[64]**]
*Veillonella parvula* (*species*)	**↑**		*p* = 0.004	171 (♀ 84; ♂ 87) SCZ: 90 Control: 81	SCZ: 28.6 Control: 32.8	[**[66]**]

**Table 5 nutrients-14-02661-t005:** Bacteria (with taxonomic level) were found in significantly higher (↑) or lower (↓) abundance in patients suffering from Alzheimer’s disease (AD). Arrows of the significant results in at least one study are represented with a grey background (*p* < 0.05). “-” in the significance column indicate that no statements regarding the significance of the results were made in the corresponding studies. Empty cells mean that no significant differences could be identified. For the studies’ demographic characteristics, sample size and mean age were included. Sample size numbers are split into females and males, as well as patients and healthy controls. The mean age is given for the patient group and the healthy control group, as a further breakdown for the gender groups was not consistently provided. The relevant source is shown in bold numbers.

Bacteria in Alzheimer’s Disease Patients	Increase	Decrease	Significance	Sample Size (*n*)	Mean Age (Years)	Source
*Actinobacteria* (*phylum*)		**↓**	*p* < 0.05	50 (♀ 35; ♂ 15) AD: 25 Control: 25	AD: 71.3 Control: 69.3	[75],[**[79]**]
*Alistipes* (*genus*)	**↑**		*p* < 0.05	50 (♀ 35; ♂ 15) AD: 25 Control: 25	AD: 71.3 Control: 69.3	[**[79]**],[80]
*Bacillus subtilis* (*species*)	↑		-	-	-	[**[76]**]
*Bacteroidaceae* (*family*)	**↑**		*p* < 0.05	50 (♀ 35; ♂ 15) AD: 25 Control: 25	AD: 71.3 Control: 69.3	[**[79]**]
*Bacteroides* (*genus*)	↑		-	-	-	[**[80]**]
*Bacteroides/Bacillus fragilis* (*species*)		↓	-	-	-	[**[74]**],[76]
*Bacteroidetes* (*phylum*)	**↑**		*p* < 0.05	50 (♀ 35; ♂ 15) AD: 25 Control: 25	AD: 71.3 Control: 69.3	[74,76,78],[**[79]**],[80]
*Bifidobacteriaceae* (*family*)		**↓**	*p* < 0.05	50 (♀ 35; ♂ 15) AD: 25 Control: 25	AD: 71.3 Control: 69.3	[**[79]**],[80]
*Bifidobacterium* (*genus*)		**↓**	*p* < 0.05	50 (♀ 35; ♂ 15) AD: 25 Control: 25	AD: 71.3 Control: 69.3	[74,76],[**[79]**],[80]
*Clostridiaceae* (*family*)		**↓**	*p* < 0.05	50 (♀ 35; ♂ 15) AD: 25 Control: 25	AD: 71.3 Control: 69.3	[**[79]**]
*Clostridium* (*genus*)		**↓**	*p* < 0.05	50 (♀ 35; ♂ 15) AD: 25 Control: 25	AD: 71.3 Control: 69.3	[**[79]**]
*Dialister* (*genus*)		**↓**	*p* < 0.05	50 (♀ 35; ♂ 15) AD: 25 Control: 25	AD: 71.3 Control: 69.3	[**[79]**]
*Escherichia* (*genus*)	**↑**		*p* < 0.001	83 (♀ 44; ♂ 39) AD: 73 Control: 10	AD: 70.5 Control: 68	[76],[**[81]**]
*Escherichia coli* (*species*)	↑		-	-	-	[**[76]**]
*Eubacterium hallii* (*species*)		↓	Not significant	83 (♀ 44; ♂ 39) AD: 73 Control: 10	AD: 70.5 Control: 68	[74],[**[81]**]
*Eubacterium rectale* (*species*)		**↓**	*p* < 0.001	83 (♀ 44; ♂ 39) AD: 73 Control: 10	AD: 70.5 Control: 68	[74,76],[**[81]**]
*Faecalibacterium prausnitzii* (*species*)		↓	Not significant	83 (♀ 44; ♂ 39) AD: 73 Control: 10	AD: 70.5 Control: 68	[74],[**[81]**]
*Firmicutes* (*phylum*)		**↓**	*p* < 0.05	50 (♀ 35; ♂ 15) AD: 25 Control: 25	AD: 71.3 Control: 69.3	[74,76],[**[79]**]
*Fusobacteriaceae* (*family*)		↓	-	-	-	[**[76]**]
*Prevotellaceae* (*family*)	↑		-	-	-	[**[76]**]

**Table 6 nutrients-14-02661-t006:** Bacteria (with taxonomic level) were found in higher (↑) or lower (↓) abundance in patients suffering from Parkinson’s disease (PD). Arrows of the significant results in at least one study are represented with a grey background (*p* < 0.05 or LDA values > 2). “-” in the significance column indicate that no statements regarding the significance of the results were made in the corresponding studies. Empty cells mean that no significant differences could be identified. For the studies’ demographic characteristics, sample size and mean age were included. Sample size numbers are split into females and males for patients and healthy controls. The mean age is given for the patient group and the healthy control group, as a further breakdown for the gender groups was not consistently provided. The relevant source is shown in bold numbers.

Bacteria in Parkinson’s Disease Patients	Increase	Decrease	Significance	Sample Size (*n*)	Mean Age (Years)	Source
*Actinobacteria* (*phylum*)	**↑**		*p* < 0.001	38 (♀ 16; ♂ 22) PD: 24 Control: 14	PD: 73.75 Control: 74.64	[80],[**[89]**]
*Akkermansia* (*genus*)	**↑**		*p* = 0.0001	327 (♀ 144; ♂ 183) PD: 197 Control: 130	PD: 68.4 Control: 70.3	[51,83,84,86,88],[**[90]**]
*Anaerotruncus* (*genus*)	**↑**		*p* = 0.047	90 (♀ 45; ♂ 45) PD: 45 Control: 45	PD: 68.1 Control: 67.9	[51],[**[91]**]
*Aquabacterium* (*genus*)	**↑**		*p* < 0.0001	90 (♀ 45; ♂ 45) PD: 45 Control: 45	PD: 68.1 Control: 67.9	[51],[**[91]**]
*Bacteroides* (*genus*)	**↑**		*p* = 0.05	72 (♀ 30; ♂ 42) PD: 38 Control: 34	PD: 61.6 Control: 45.1	[83],[**[92]**]
*Bacteroidetes* (*phylum*)		**↓**	*p* = 0.045	38 (♀ 16; ♂ 22) PD: 24 Control: 14	PD: 73.75 Control: 74.64	[84],[**[89]**],[93]
*Bifidobacteriaceae* (*family*)	**↑**		*p* < 0.0001	327 (♀ 144; ♂ 183) PD: 197 Control: 130	PD: 68.4 Control: 70.3	[84],[**[90]**]
*Blautia* (*genus*)		**↓**	*p* = 0.018	38 (♀ 16; ♂ 22) PD: 24 Control: 14	PD: 73.75 Control: 74.64	[51],[**[89]**]
*Butyricicoccus* (*genus*)	**↑**		*p* = 0.034	90 (♀ 45; ♂ 45) PD: 45 Control: 45	PD: 68.1 Control: 67.9	[51],[**[91]**]
*Christensenellaceae* (*family*)	**↑**		*p* < 0.0001	327 (♀ 144; ♂ 183) PD: 197 Control: 130	PD: 68.4 Control: 70.3	[84],[**[90]**]
*Clostridium IV* (*genus*)	**↑**		*p* < 0.0001	90 (♀ 45; ♂ 45) PD: 45 Control: 45	PD: 68.1 Control: 67.9	[51],[**[91]**]
*Clostridium XVIII* (*genus*)	**↑**		*p* = 0.03	90 (♀ 45; ♂ 45) PD: 45 Control: 45	PD: 68.1 Control: 67.9	[51],[**[91]**]
*Coprococcus* (*genus*)		**↓**	*p* = 0.03	72 (♀ 30; ♂ 42) PD: 38 Control: 34	PD: 61.6 Control: 45.1	[51],[**[92]**]
*Enterococcaceae* (*family*)		↓	-	68 (♀ 26; ♂ 42) PD: 34 Control: 34	PD: 67.7 Control: 64.6	[51],[**[93]**]
*Enterococcus* (*genus*)	**↑**		*p* = 0.006	38 (♀ 16; ♂ 22) PD: 24 Control: 14	PD: 73.75 Control: 74.64	[51],[**[89]**]
*Escherichia-Shigella* (*genus*)	**↑**		*p* = 0.038	38 (♀ 16; ♂ 22) PD: 24 Control: 14	PD: 73.75 Control: 74.64	[51],[**[89]**]
*Faecalibacterium* (*genus*)		**↓**	*p* < 0.05	327 (♀ 144; ♂ 183) PD: 197 Control: 130	PD: 68.4 Control: 70.3	[51,84,86,88],[**[90]**]
*Firmicutes* (*phylum*)		**↓**	*p* = 0.03	72 (♀ 30; ♂ 42) PD: 38 Control: 34	PD: 61.6 Control: 45.1	[84],[**[92]**]
*Holdemania* (*genus*)	**↑**		*p* = 0.004	90 (♀ 45; ♂ 45) PD: 45 Control: 45	PD: 68.1 Control: 67.9	[51],[**[91]**]
*Lachnospiraceae* (*family*)		**↓**	*p* = 0.02	72 (♀ 30; ♂ 42) PD: 38 Control: 34	PD: 61.6 Control: 45.1	[84,86,88],[**[92]**]
*Lactobacillaceae* (*family*)	**↑**		*p* < 0.0001	327 (♀ 144; ♂ 183) PD: 197 Control: 130	PD: 68.4 Control: 70.3	[83,84,85,87],[**[90]**]
*Lactobacillus* (*genus*)	**↑**		*p* < 0.0001	327 (♀ 144; ♂ 183) PD: 197 Control: 130	PD: 68.4 Control: 70.3	[84],[**[90]**]
*Lactobacillus* (*genus*)		**↓**	LDA > 2	90 (♀ 45; ♂ 45) PD: 45 Control: 45	PD: 68.1 Control: 67.9	[51,85],[**[91]**]
*Prevotella* (*genus*)		↓	*p* = 0.28	88 (♀ 46; ♂ 42) PD: 52 Control: 36	PD: 68.9 Control: 68.4	[51,84,85,86,88],[**[94]**]
*Prevotellaceae* (*family*)		↓	Not significant	38 (♀ 16; ♂ 22) PD: 24 Control: 14	PD: 73.75 Control: 74.64	[84],[**[89]**],[93]
*Proteus* (*genus*)	**↑**		*p* = 0.022	38 (♀ 16; ♂ 22) PD: 24 Control: 14	PD: 73.75 Control: 74.64	[51,85],[**[89]**]
*Roseburia* (*genus*)		**↓**	*p* < 0.05	327 (♀ 144; ♂ 183) PD: 197 Control: 130	PD: 68.4 Control: 70.3	[51,83],[**[90]**]
*Ruminococcaceae* (*family*)	**↑**		*p* < 0.05	20 (♀ 8; ♂ 12) PD: 10 Control: 10	PD: 79.5 Control: 76.5	[84],[**[95]**]
*Ruminococcus* (*species*)		**↓**	*p* = 0.019	38 (♀ 16; ♂ 22) PD: 24 Control: 14	PD: 73.75 Control: 74.64	[**[89]**]
*Sediminibacterium* (*genus*)		**↓**	LDA > 2	90 (♀ 45; ♂ 45) PD: 45 Control: 45	PD: 68.1 Control: 67.9	[51],[**[91]**]
*Sphingomonas* (*genus*)	**↑**		*p* < 0.05	90 (♀ 45; ♂ 45) PD: 45 Control: 45	PD: 68.1 Control: 67.9	[51],[**[91]**]
*Streptococcus* (*genus*)	**↑**		*p* = 0.01	38 (♀ 16; ♂ 22) PD: 24 Control: 14	PD: 73.75 Control: 74.64	[51],[**[89]**]
*Verrucomicrobiaceae* (*family*)	**↑**		*p* = 0.05	72 (♀ 30; ♂ 42) PD: 38 Control: 34	PD: 61.6 Control: 45.1	[51],[**[92]**]

**Table 7 nutrients-14-02661-t007:** Bacteria (with taxonomic level) were found in higher (↑) or lower (↓) abundance in patients suffering from major depressive disorder (MDD). Arrows of the significant results in at least one study are represented with a grey background (*p* < 0.05 or LDA values > 2). “-” in the significance column indicate that no statements regarding the significance of the results were made in the corresponding studies. Empty cells mean that no significant differences could be identified. For the studies’ demographic characteristics, sample size and mean age were included. Sample size numbers are split into females and males for patients and healthy controls. The mean age is given for the patient group and the healthy control group, as a further breakdown for the gender groups was not consistently provided. The relevant source is shown in bold numbers.

Bacteria in Major Depressive Disorder Patients	Increase	Decrease	Significance	Sample Size (*n*)	Mean Age (Years)	Source
*Actinobacteria* (*phylum*)	**↑**		*p* < 0.05	76 (♀ 34; ♂ 42) MDD: 46 Control: 30	MDD: 26.2 Control: 26.8	[102],[**[105]**]
*Actinomycetaceae* (*family*)	↑		-	-	-	[100],[**[104]**]
*Alistipes* (*genus*)	**↑**		*p* < 0.05	76 (♀ 34; ♂ 42) MDD: 46 Control: 30	MDD: 26.2 Control: 26.8	[51,100,102],[**[105]**]
*Atopobium* (*genus*)	↑		-	-	-	[**[99]**],[100]
*Bacteroides* (*genus*)	**↑**		*p* = 0.007	-	-	[102],[**[106]**]
*Bacteroidetes* (*phylum*)	**↑**		*p* < 0.05	76 (♀ 34; ♂ 42) MDD: 46 Control: 30	MDD: 26.2 Control: 26.8	[101,103],[**[105]**]
*Bifidobacteriaceae* (*family*)	**↑**		*p* = 0.004	382 (♀ 228; ♂ 154) MDD: 165 Control: 217	MDD: 45.1 Control: 36.1	[102],[**[107]**]
*Bifidobacterium* (*genus*)	**↑**		*p* < 0.01	61 (♀ 38; ♂ 23) MDD: 31 Control: 30	MDD: 41.58 Control: 39.47	[99,102],[**[108]**]
*Bifidobacterium* (*genus*)		**↓**	*p* = 0.012	100 (♀ 53; ♂ 47) MDD: 43 Control: 57	MDD: 39.4 Control: 42.8	[51,100,103,104],[**[109]**]
*Blautia* (*genus*)	**↑**		*p* < 0.05	76 (♀ 34; ♂ 42) MDD: 46 Control: 30	MDD: 26.2 Control: 26.8	[100],[**[105]**]
*Christensenellaceae* (*family*)		**↓**	*p* = 0.0395	90 (♀ 72; ♂ 18) MDD: 43 Control: 47	MDD: 21.9 Control: 22.1	[102],[**[110]**]
*Clostridium* (*genus*)	**↑**		*p* < 0.01	61 (♀ 38; ♂ 23) MDD: 31 Control: 30	MDD: 41.58 Control: 39.47	[100],[**[108]**]
*Coprococcus* (*genus*)		↓	*p* = 0.101	121 (♀ 76; ♂ 45) MDD: 58 Control: 63	MDD: 40.6 Control: 41.8	[100,102,104],[**[111]**]
*Dialister* (*genus*)		**↓**	*p* < 0.05	76 (♀ 34; ♂ 42) MDD: 46 Control: 30	MDD: 26.2 Control: 26.8	[99],[**[105]**]
*Eggerthella* (*genus*)	**↑**		*p* < 0.01	61 (♀ 38; ♂ 23) MDD: 31 Control: 30	MDD: 41.58 Control: 39.47	[99],[**[108]**]
*Enterobacteriaceae* (*family*)	**↑**		*p* < 0.05	76 (♀ 34; ♂ 42) MDD: 46 Control: 30	MDD: 26.2 Control: 26.8	[51],[**[105]**]
*Escherichia* (*genus*)		↓	-	-	-	[100],[**[104]**]
*Eubacterium* (*genus*)	↑		*p* = 0.065	121 (♀ 76; ♂ 45) MDD: 58 Control: 63	MDD: 40.6 Control: 41.8	[**[111]**]
*Eubacterium rectale* (*species*)	**↑**		*p* < 0.01	61 (♀ 38; ♂ 23) MDD: 31 Control: 30	MDD: 41.58 Control: 39.47	[**[108]**]
*Faecalibacterium* (*genus*)		**↓**	*p* < 0.05	76 (♀ 34; ♂ 42) MDD: 46 Control: 30	MDD: 26.2 Control: 26.8	[51,99,100,102,103,104],[**[105]**]
*Firmicutes* (*phylum*)		**↓**	*p* < 0.05	76 (♀ 34; ♂ 42) MDD: 46 Control: 30	MDD: 26.2 Control: 26.8	[101,103],[**[105]**]
*Flavonifractor* (*genus*)	**↑**		LDA > 2	76 (♀ 34; ♂ 42) MDD: 46 Control: 30	MDD: 26.2 Control: 26.8	[102],[**[105]**]
*Lactobacillus* (*genus*)	**↑**		*p* < 0.01	61 (♀ 38; ♂ 23) MDD: 31 Control: 30	MDD: 41.58 Control: 39.47	[**[108]**]
*Oscillibacter* (*genus*)	**↑**		*p* < 0.05	76 (♀ 34; ♂ 42) MDD: 46 Control: 30	MDD: 26.2 Control: 26.8	[100],[**[105]**]
*Parabacteroides* (*genus*)	**↑**		*p* < 0.05	76 (♀ 34; ♂ 42) MDD: 46 Control: 30	MDD: 26.2 Control: 26.8	[102,105]
*Paraprevotella* (*genus*)	**↑**		*p* = 0.041	67 (♀ 27; ♂ 40) MDD: 34 Control: 33	MDD: 45.8 Control: 45.8	[100,104],[**[112]**]
*Prevotella* (*genus*)	**↑**		*p* < 0.01	61 (♀ 38; ♂ 23) MDD: 31 Control: 30	MDD: 41.58 Control: 39.47	[103],[**[108]**]
*Prevotellaceae* (*family*)		**↓**	*p* < 0.05	76 (♀ 34; ♂ 42) MDD: 46 Control: 30	MDD: 26.2 Control: 26.8	[100,103,104],[**[105]**]
*Proteobacteria* (*phylum*)	**↑**		*p* < 0.05	76 (♀ 34; ♂ 42) MDD: 46 Control: 30	MDD: 26.2 Control: 26.8	[103],[**[105]**]
*Roseburia* (*genus*)	**↑**		*p* < 0.05	76 (♀ 34; ♂ 42) MDD: 46 Control: 30	MDD: 26.2 Control: 26.8	[**[105]**],[111]
*Ruminococcaceae* (*family*)		**↓**	*p* < 0.05	76 (♀ 34; ♂ 42) MDD: 46 Control: 30	MDD: 26.2 Control: 26.8	[102],[**[105]**]
*Ruminococcus* (*genus*)		**↓**	*p* < 0.05	76 (♀ 34; ♂ 42) MDD: 46 Control: 30	MDD: 26.2 Control: 26.8	[100,103,104],[**[105]**]
*Streptococcus* (*genus*)	**↑**		*p* < 0.01	61 (♀ 38; ♂ 23) MDD: 31 Control: 30	MDD: 41.58 Control: 39.47	[102],[**[108]**],[111,113]
*Sutterella* (*genus*)		↓	-	73 (♀ 51; ♂ 22) MDD: 36 Control: 37	MDD: 45.83 Control: 41.19	[102],[**[113]**]
*Sutterellaceae* (*family*)		↓	-	-	-	[100,102],[**[104]**]
*Veillonellaceae* (*family*)		**↓**	*p* < 0.05	76 (♀ 34; ♂ 42) MDD: 46 Control: 30	MDD: 26.2 Control: 26.8	[100,104],[**[105]**]

**Table 8 nutrients-14-02661-t008:** Bacteria (with taxonomic level) were found in higher (↑) or lower (↓) abundance in patients suffering from bipolar disease (BD). Arrows of the significant results in at least one study are represented with a grey background (*p* < 0.05 or LDA values > 2). Empty cells mean that no significant differences could be identified. For the studies’ demographic characteristics, sample size and mean age were included. Sample size numbers are split into females and males for patients and healthy controls. The mean age is given for the patient group and the healthy control group, as a further breakdown for the gender groups was not consistently provided. The relevant source is shown in bold numbers.

Bacteria in Bipolar Disorder Patients	Increase	Decrease	Significance	Sample Size (*n*)	Mean Age (Years)	Source
*Actinobacteria* (*phylum*)	**↑**		*p* < 0.01	60 (♀ 31; ♂ 29) BD: 30 Control: 30	BD: 38.40 Control: 39.47	[102],[**[108]**],[116,118]
*Atopobium Cluster* (*genus*)	**↑**		*p* < 0.001	63 (♀ 27; ♂ 36) BD: 36 Control: 27	BD: 32.64 Control: 28.89	[115,116],[**[119]**]
*Bacteroides* (*genus*)		**↓**	*p* < 0.01	60 (♀ 31; ♂ 29) BD: 30 Control: 30	BD: 38.40 Control: 39.47	[**[108]**],[116]
*Bacteroides* (*genus*)	**↑**		*p* < 0.05	97 (♀ 47; ♂ 50) BD: 52 Control: 45	BD: 24.15 Control: 36.29	[116],[**[120]**]
*Bacteroidetes* (*phylum*)	**↑**		*p* < 0.05	97 (♀ 47; ♂ 50) BD: 52 Control: 45	BD: 24.15 Control: 36.29	[116],[**[120]**]
*Bacteroidetes* (*phylum*)		**↓**	*p* < 0.01	60 (♀ 31; ♂ 29) BD: 30 Control: 30	BD: 38.40 Control: 39.47	[**[108]**]
*Bifidobacterium* (*genus*)	**↑**		*p* < 0.01	60 (♀ 31; ♂ 29) BD: 30 Control: 30	BD: 38.40 Control: 39.47	[**[108]**],[116]
*Clostridium Cluster IV* (*genus*)	**↑**		*p* < 0.001	63 (♀ 27; ♂ 36) BD: 36 Control: 27	BD: 32.64 Control: 28.89	[115,116],[**[119]**]
*Clostridium* (*genus*)	**↑**		*p* < 0.01	60 (♀ 31; ♂ 29) BD: 30 Control: 30	BD: 38.40 Control: 39.47	[**[108]**],[116]
*Coprococcus* (*genus*)		**↓**	*p* < 0.05	97 (♀ 47; ♂ 50) BD: 52 Control: 45	BD: 24.15 Control: 36.29	[115,116],[**[120]**]
*Desulfovibrio* (*genus*)	**↑**		*p* < 0.01	60 (♀ 31; ♂ 29) BD: 30 Control: 30	BD: 38.40 Control: 39.47	[**[108]**],[116]
*Enterobacter* (*genus*)	**↑**		*p* < 0.001	63 (♀ 27; ♂ 36) BD: 36 Control: 27	BD: 32.64 Control: 28.89	[115,116],[**[119]**]
*Escherichia* (*genus*)	**↑**		*p* < 0.01	60 (♀ 31; ♂ 29) BD: 30 Control: 30	BD: 38.40 Control: 39.47	[**[108]**],[115,116]
*Faecalibacterium* (*genus*)		**↓**	*p* < 0.05	97 (♀ 47; ♂ 50) BD: 52 Control: 45	BD: 24.15 Control: 36.29	[115,116,117,118],[**[120]**]
*Faecalibacterium prausnitzii* (*species*)	**↑**		*p* = 0.030	63 (♀ 27; ♂ 36) BD: 36 Control: 27	BD: 32.64 Control: 28.89	[115,116],[**[119]**]
*Firmicutes* (*phylum*)	**↑**		*p* < 0.01	60 (♀ 31; ♂ 29) BD: 30 Control: 30	BD: 38.40 Control: 39.47	[**[108]**]
*Flavonifractor* (*genus*)	**↑**		*p* < 0.05	190 (♀ 117; ♂ 73) BD: 113 Control: 77	BD: 31 Control: 28	[115,116],[**[121]**]
*Halomonas* (*genus*)	**↑**		*p* < 0.05	97 (♀ 47; ♂ 50) BD: 52 Control: 45	BD: 24.15 Control: 36.29	[116],[**[120]**]
*Klebsiella* (*genus*)	**↑**		*p* < 0.05	60 (♀ 31; ♂ 29) BD: 30 Control: 30	BD: 38.40 Control: 39.47	[**[108]**],[115,116]
*Oscillibacter* (*genus*)	**↑**		*p* < 0.01	60 (♀ 31; ♂ 29) BD: 30 Control: 30	BD: 38.40 Control: 39.47	[**[108]**],[116]
*Parabacteroides* (*genus*)	**↑**		*p* < 0.05	97 (♀ 47; ♂ 50) BD: 52 Control: 45	BD: 24.15 Control: 36.29	[116],[**[120]**]
*Prevotella* (*genus*)		↓	Not significant	60 (♀ 31; ♂ 29) BD: 30 Control: 30	BD: 38.40 Control: 39.47	[**[108]**]
*Proteobacteria* (*phylum*)	**↑**		*p* < 0.01	60 (♀ 31; ♂ 29) BD: 30 Control: 30	BD: 38.40 Control: 39.47	[102],[**[108]**]
*Roseburia* (*genus*)		**↓**	*p* < 0.05	97 (♀ 47; ♂ 50) BD: 52 Control: 45	BD: 24.15 Control: 36.29	[115,116],[**[120]**]
*Ruminococcaceae* (*family*)		**↓**	LDA > 2	97 (♀ 47; ♂ 50) BD: 52 Control: 45	BD: 24.15 Control: 36.29	[115,116,117,118],[**[120]**]
*Streptococcus* (*genus*)	**↑**		*p* < 0.01	60 (♀ 31; ♂ 29) BD: 30 Control: 30	BD: 38.40 Control: 39.47	[**[108]**],[116]

**Table 9 nutrients-14-02661-t009:** Arrangement of the data collected for the individual disorders, allowing to compare the changes in bacteria strains found. The list contains the strains of bacteria along with their taxonomic level in brackets, where an overlap could be observed. An increase in abundance is depicted by light green cells containing “↑”. Light red cells with “↓” mark a decrease in the corresponding bacteria strain. “↑↓” means that data pointing in both directions could be found. The bacterial strains were included in this table regardless of the level of significance in the original literature. For the sources and levels of significance, please consult the corresponding bacteria tables in the previous chapters. ADHD = attention deficit hyperactivity disorder; ASD = autism spectrum disorder; AD = Alzheimer’s disease; PD = Parkinson’s disease; MDD = major depressive disorder; BD = bipolar disorder.

	ADHD	ASD	Schizophrenia	AD	PD	MDD	BD
*Actinobacteria* (*phylum*)	↑	↑	↑	↓	↑	↑	↑
*Alistipes* (*genus*)		↑↓		↑		↑	
*Atopobium* (*genus*)						↑	↑
*Bacteroides* (*genus*)	↑	↑		↑	↑	↑	↑↓
*Bacteroidetes* (*phylum*)		↑↓		↑	↓	↑	↑↓
*Bifidobacteriaceae* (*family*)				↓	↑	↑	
*Bifidobacterium* (*genus*)	↑	↓	↑↓	↓		↑↓	↑
*Blautia* (*genus*)					↓	↑	
*Christensenellaceae* (*family*)					↑	↓	
*Clostridium* (*genus*)		↑	↓	↓		↑	↑
*Coprococcus* (*genus*)		↓			↓	↓	↓
*Desulfovibrio* (*genus*)		↑					↑
*Dialister* (*genus*)	↓	↓		↓		↓	
*Eggerthella* (*genus*)			↑			↑	
*Enterobacteriaceae* (*family*)		↑				↑	
*Enterococcaceae* (*family*)			↓		↓		
*Enterococcus* (*genus*)		↓	↓		↑		
*Escherichia* (*genus*)				↑		↓	↑
*Escherichia coli* (*species*)		↓	↓	↑			
*Eubacterium* (*genus*)		↓				↑	
*Eubacterium rectale* (*species*)				↓		↑	
*Faecalibacterium* (*genus*)	↓	↑			↓	↓	↓
*Faecalibacterium prausnitzii* (*species*)				↓			↑
*Firmicutes* (*phylum*)	↓	↑↓		↓	↓	↓	↑
*Flavonifractor* (*genus*)						↑	↑
*Lachnospiraceae* (*family*)		↑↓			↓		
*Lactobacillaceae* (*family*)		↑			↑		
*Lactobacillus* (*genus*)	↓	↑↓	↑		↑↓	↑	
*Neisseria* (*genus*)	↑	↓					
*Oscillibacter* (*genus*)						↑	↑
*Parabacteroides* (*genus*)	↓	↑↓				↑	↑
*Prevotella* (*genus*)	↓	↓			↓	↑	↓
*Prevotellaceae* (*family*)				↑	↓	↓	
*Proteobacteria* (*phylum*)			↓			↑	↑
*Roseburia* (*genus*)		↑			↓	↑	↓
*Ruminococcaceae* (*family*)		↓			↑	↓	↓
*Ruminococcus* (*genus*)		↑			↓	↓	
*Streptococcus* (*genus*)		↓			↑	↑	↑
*Sutterella* (*genus*)		↑↓	↓			↓	
*Sutterellaceae* (*family*)		↑				↓	
*Veillonellaceae* (*family*)		↑				↓	

**Table 10 nutrients-14-02661-t010:** A compilation of the literature on SCFA level aberrations in the different diseases when compared to healthy individuals. “↑” on a light green background symbolises an increase in the corresponding SCFA. A light red background with “↓” stands for a decrease, respectively. Cells containing “↑↓” imply that the findings were inconsistent and changes in patients suffering from the disease went in both directions. If no literature could be found, the cells remained empty. ADHD = attention deficit hyperactivity disorder; ASD = autism spectrum disorder; AD = Alzheimer’s disease; PD = Parkinson’s disease; MDD = major depressive disorder; BD = bipolar disorder.

	ADHD	ASD	Schizophrenia	AD	PD	MDD	BD
General SCFA levels	↓ [27]	↑ [3,29,45]	↑ [68]	↓ [71,123]	↓ [29,51,68,124]	↓ [29]	
		↓ [29,44,46]					
Butyrate		↑↓ [29]			↓ [29,68,124]		↓ [115,116]
Acetate		↑↓ [29]	↑ [68]	↓ [125]	↓ [29,68]	↓ [29]	
Valerate		↑↓ [29]		↓ [125]			
Isovaleric Acid		↑ [29]		↓ [125]		↓ [29]	
Propionate		↑↓ [29]	↑ [68]	↓ [125]	↓ [29,68,124]	↓ [29]	
		↑ [68]					
Isocaproic acid						↑ [29]	
Isobutyric acid		↑ [29]					

## Data Availability

Not applicable.

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
