# Peer review of "Overlapping Mechanisms of Action of Brain-Active Bacteria and Bacterial Metabolites in the Pathogenesis of Common Brain Diseases"

_nutrients, 2022, doi:10.3390/nu14132661_

Round 1

Reviewer 1 Report

In this manuscript, the authors summarized the contribution of the microbiota-gut-brain-axis to the pathophysiology of seven neurological and psychiatric disorders. They further compared the changed microbiota populations and potential brain-gut signaling molecules among the disorders and identified some common pattern among those disorders. The synthetized data could potentially help identify novel treatment options. The following points can be further addressed before publication.

1. A flow chart may be generated to give an overview of the number of studies included and excluded

2. Is the manuscript prepared according to the PRISMA preferred reporting items for systemic review?

3. In the table of summary for each disease (e.g. table 2 and following tables), it could be more informative to include a column that display the number study subjects in each study and their demographic information such as age sex and other relevant information of the study subjects from the reference if available. Discuss the influence of age, sex and other relevant demographic information may have on the gut microbiota population

4. Please summarized the limitations of the current systemic review.

Author Response

Dear Editors,                                                                                   June 21, 2022

Thank you very much for reviewing the manuscript nutrients-1770244, entitled: "Overlapping Mechanisms of Action of Brain-Active Bacteria and Bacterial Metabolites in the Pathogenesis of Common Brain Diseases” by T. P. Eicher and M.H. Mohajeri.

We are specifically thankful for this Reviewer’s comment that “The synthetized data could potentially help identify novel treatment options”.

We have now revised this manuscript according to all suggestions made by the reviewers and have added the additional data and citations requested. All changes are indicated in the text by yellow colour for better visibility and transparency. In this letter, our responses follow each reviewer’s comment.  We would like to draw the attention of the journal editors to the last comment of reviewer 2. In compliance with her/his comment, we are also submitting the mentioned figures as separate files with high resolution. For completeness, copies of these figures are included in the manuscript, nonetheless. 

Reviewer 1

1- A flow chart may be generated to give an overview of the number of studies included and excluded.

Thank you for your remark. We have clearly described in the section Materials and Methods the inclusion and exclusion criteria for reviewed literature. As this is a rather sizeable manuscript, with 60 pages including 10 large tables and 13 extensive figures, we did not deem it appropriate to add yet another figure, which would largely replicate the inclusion/exclusion criteria. We hope that the reviewer kindly accepts this explanation for not representing such a flow chart.

2- Is the manuscript prepared according to the PRISMA preferred reporting items for systemic review?

We appreciate this question. Even if we have tried to adhere to the PRISMA guiding instructions, we have not prepared this manuscript as a systematic review. Indeed, we have nowhere in the text or in the communication to the journal raised a claim that our manuscript will be representing a “systematic review”.

3- In the table of summary for each disease (e.g. table 2 and following tables), it could be more informative to include a column that display the number study subjects in each study and their demographic information such as age sex and other relevant information of the study subjects from the reference if available. Discuss the influence of age, sex and other relevant demographic information may have on the gut microbiota population.

We have now included the information, if available, in tables 2-8, as requested. In particular, sample size, age, and gender are given for females and males for patients and healthy controls. This information can be found in the figure legends in separate columns in tables 2-8 in yellow colour (pages, 8, 9, 12, 13, 14, 15, 20, 21, 22, 23, 27, 28, 32, 33, 34, 38, 39, 40, 41, 45, 46, 47.

4- Please summarized the limitations of the current systemic review.

We appreciate this comment. A section on the limitation of this study is now added on page 53, lines 1263 to 1276.

We hope that you will find the revised version of our manuscript acceptable for publication.

Sincerely yours,

MH Mohajeri, PhD

Reviewer 2 Report

The review of Eicher and Mohajeri deals with the involvement of the microbiota and the bacteria metabolites in several brain diseases including neurodegenerative disorders. The topic is of interest and although several reviews have been published on the subject, the present work has the merit of being extremely complete, detailed and also discussing some less treated pathologies. The databases and keywords used are valuable and guarantee an updated version of the bibliography on the topic. The paper is of interest to the wide readership of Nutrients.

In table 1 report the appropriate reference number in each line so to facilitate the search by the reader of the works of interest.

Figure-2-13 are of sure interest but they are not readable. Fonts must be larger or authors should find another way to present the patterns.

Author Response

Dear Editors,                                                                               June 21, 2022

Thank you very much for reviewing the manuscript nutrients-1770244, entitled: "Overlapping Mechanisms of Action of Brain-Active Bacteria and Bacterial Metabolites in the Pathogenesis of Common Brain Diseases” by T. P. Eicher and M.H. Mohajeri.

We are specifically thankful for the comments that “The topic is of interest and although several reviews have been published on the subject, the present work has the merit of being extremely complete, detailed and also discussing some less treated pathologies. The databases and keywords used are valuable and guarantee an updated version of the bibliography on the topic. The paper is of interest to the wide readership of Nutrients”.

We have now revised this manuscript according to all suggestions made by the reviewers and have added the additional data and citations requested. All changes are indicated in the text by yellow colour for better visibility and transparency. In this letter, our responses follow each reviewer’s comment.  We would like to draw the attention of the journal editors to the last comment of the reviewer 2. In compliance with her/his comment, we are also submitting the mentioned figures as separate files with high resolution. For completeness, copies of these figures are included in the manuscript, nonetheless.

Reviewer 2

- In table 1 report the appropriate reference number in each line so to facilitate the search by the reader of the works of interest.

Thank you for your remark. We have complied with this request and have added the literature references to table 1. This information is highlighted on page 5, table 1.

-Figure-2-13 are of sure interest but they are not readable. Fonts must be larger or authors should find another way to present the patterns.

Thank you for your comment. We have now submitted the original high-resolution figures 2-13 as separate files to the journal, in addition to including them in the main text of the manuscript.

We hope that you will find the revised version of our manuscript acceptable for publication.

Sincerely yours,

MH Mohajeri, PhD
